# Allosteric regulators selectively prevent Ca$^{2+}$-feedback of Ca$_V$ and Na$_V$ channels

**Jacqueline Niu[1], Ivy E Dick[2], Wanjun Yang[3], Moradeke A Bamgboye[2], David T Yue[1], Gordon Tomaselli[3], Takanari Inoue[4,5]\*, Manu Ben-Johny[6]\***

[1]Department of Biomedical Engineering, Johns Hopkins University, Baltimore, United States; [2]Department of Physiology, University of Maryland, Baltimore, United States; [3]Department of Cardiology, Johns Hopkins University, Baltimore, United States; [4]Department of Cell Biology, Johns Hopkins University, Baltimore, United States; [5]Center for Cell Dynamics, Institute for Basic Biomedical Sciences, Johns Hopkins University, Baltimore, United States; [6]Department of Physiology and Cellular Biophysics, College of Physicians and Surgeons, Columbia University, New York, United States

**Abstract** Calmodulin (CaM) serves as a pervasive regulatory subunit of Ca$_V$1, Ca$_V$2, and Na$_V$1 channels, exploiting a functionally conserved carboxy-tail element to afford dynamic Ca$^{2+}$-feedback of cellular excitability in neurons and cardiomyocytes. Yet this modularity counters functional adaptability, as global changes in ambient CaM indiscriminately alter its targets. Here, we demonstrate that two structurally unrelated proteins, SH3 and cysteine-rich domain (stac) and fibroblast growth factor homologous factors (fhf) selectively diminish Ca$^{2+}$/CaM-regulation of Ca$_V$1 and Na$_V$1 families, respectively. The two proteins operate on allosteric sites within upstream portions of respective channel carboxy-tails, distinct from the CaM-binding interface. Generalizing this mechanism, insertion of a short RxxK binding motif into Ca$_V$1.3 carboxy-tail confers synthetic switching of CaM regulation by Mona SH3 domain. Overall, our findings identify a general class of auxiliary proteins that modify Ca$^{2+}$/CaM signaling to individual targets allowing spatial and temporal orchestration of feedback, and outline strategies for engineering Ca$^{2+}$/CaM signaling to individual targets.

DOI: https://doi.org/10.7554/eLife.35222.001

*For correspondence:
jctinoue@jhmi.edu (TI);
mbj2124@cumc.columbia.edu
(MB-J)

**Competing interests:** The authors declare that no competing interests exist.

## Introduction

Supporting vital biological functions, voltage-gated calcium (Ca$_V$1 and Ca$_V$2) and sodium (Na$_V$1) channels are tuned by the Ca$^{2+}$-binding protein, calmodulin (CaM) (*Ben-Johny et al., 2015*; *Catterall et al., 2017*; *Saimi and Kung, 2002*). Na$_V$1 supports action potential initiation and propagation (*Hille, 2001*), while Ca$_V$1/2 initiate muscle contraction, neurotransmission, and gene transcription (*Berridge et al., 2000*; *Clapham, 2007*; *Maier and Bers, 2002*). Despite divergent functions, these channel families share a conserved carboxy-tail element, termed Ca$^{2+}$-inactivating (CI) module, that harbors CaM. Functionally, the CI module confers dynamic Ca$^{2+}$-dependent regulation to Ca$_V$1, Ca$_V$2, and Na$_V$1 that manifests as either inactivation (CDI) or facilitation (CDF), negative and positive feedback, respectively (*Ben-Johny et al., 2015*; *Minor and Findeisen, 2010*). Yet, this modularity poses a challenge – mechanisms that tune Ca$^{2+}$/CaM-feedback must distinguish between structurally similar targets. Global inhibition of CaM indiscriminately alters many processes (*Persechini and Stemmer, 2002*). Given the abundance of CaM-regulated proteins, mechanisms that adjust CaM signaling to individual targets are crucial (*Marshall et al., 2015*; *Saimi and Kung, 2002*). Physiologically, Ca$^{2+}$-regulation of Ca$_V$1 is critical for cardiac electrical stability (*Limpitikul et al., 2014*; *Mahajan et al., 2008*), rhythmicity of oscillatory neurons (*Chan et al., 2007*;

*Huang et al., 2012*), and vesicle release at ribbon synapses (*Joiner and Lee, 2015*). Ca$_V$2 modulation contributes to short-term synaptic plasticity and spatial learning (*Adams et al., 2010*; *Jackman and Regehr, 2017*; *Nanou et al., 2016*), while Na$_V$1 modulation tunes excitability of skeletal and cardiac muscle (*Pitt and Lee, 2016*; *Van Petegem et al., 2012*). Consequently, aberrant channel regulation underlies numerous maladies including cardiac arrhythmias (*Venetucci et al., 2012*; *Zimmer and Surber, 2008*), neurological and neuropsychiatric disorders (*Adams and Snutch, 2007*; *Striessnig et al., 2010*; *Zamponi, 2016*), and skeletal myotonia (*Cannon, 2015*).

Src homology 3 (SH3) and cysteine-rich domain (C1) proteins (stac) have emerged as attractive candidates that modulate Ca$_V$ gating and trafficking (*Polster et al., 2015*; *Suzuki et al., 1996*). Initially identified in association with congenital skeletal myopathies as a structural protein that abets Ca$_V$1.1 plasmalemmal trafficking (*Horstick et al., 2013*; *Linsley et al., 2017c*; *Polster et al., 2015*), stac also suppresses Ca$_V$1.2 CDI (*Campiglio et al., 2018*; *Wong King Yuen et al., 2017*). Even so, the specificity of stac in tuning Ca$^{2+}$-regulation of the broader Ca$_V$/Na$_V$ family, the underlying elementary mechanisms, and molecular determinants remain to be fully elucidated (*Wong King Yuen et al., 2017*). Stac isoforms (stac1-3) share a common architecture containing a C1 and two SH3 domains fused via a linker, and exhibit tissue-specific expression (*Suzuki et al., 1996*). Stac1/2 are expressed throughout the brain (*Nelson et al., 2013*; *Suzuki et al., 1996*), the peripheral nervous system (*Legha et al., 2010*), the retina (*Wilhelm et al., 2014*), and the inner ear (*Cai et al., 2015*), while stac3 is limited to the skeletal muscle (*Nelson et al., 2013*). Resolving mechanisms by which stac modulates Ca$_V$ may furnish long-sought physiological insights (*Suzuki et al., 1996*).

Evolutionarily distinct from stac, fibroblast growth factor (fgf) homologous factors (fhf1-4, fgf11-14) are unconventional fgf that lack a secretory signal and serve as intracellular regulators of Na$_V$ gating and trafficking (*Goldfarb, 2005*; *Pablo and Pitt, 2016*). Curiously, fhf interacts with the Na$_V$ CI module in close proximity to the CaM binding interface, suggesting interplay between these modulators (*Wang et al., 2012*; *Wang et al., 2011b*). Yet, functionally, fhf is thought to modulate only voltage-dependent fast inactivation (*Goldfarb et al., 2007*; *Lou et al., 2005*; *Wang et al., 2011a*), with changes in Ca$^{2+}$-regulation unknown. Fhf isoforms are differentially expressed in the brain (*Smallwood et al., 1996*; *Yan et al., 2014*), peripheral nervous system (*Ornitz and Itoh, 2001*), and cardiac (*Wei et al., 2011*) and skeletal muscle (*Kraner et al., 2012*). Genetic variation in fhf4 is linked to spinocerebellar ataxia 27 (*Coebergh et al., 2014*) and fhf1 to cardiac arrhythmias (*Wei et al., 2011*), hinting at their relevance for regulating neuronal and cardiac excitability.

By leveraging synergistic insights from Ca$_V$ and Na$_V$ channels, we demonstrate that stac selectively diminishes Ca$^{2+}$-regulation of Ca$_V$1. In-depth analysis shows that stac binds to a distinct channel interface from CaM and uses an allosteric mechanism to lock Ca$_V$1 into a high open probability ($P_O$) gating mode. We further localize a minimal motif that recapitulates stac modulation of Ca$_V$1 gating. Paralleling stac-effect on Ca$_V$1, fhf reduces CDI of Na$_V$1 with no effect on Ca$_V$1. In all, our findings point to a general class of auxiliary proteins that intercept CaM signaling to individual targets, allowing spatial and temporal orchestration of Ca$^{2+}$-feedback.

## Results

### Stac selectively suppresses Ca$^{2+}$-feedback of Ca$_V$1 channels

We sought to determine stac effect on Ca$_V$1, Ca$_V$2, and Na$_V$1 channels in heterologous systems. *Figure 1A* shows baseline effects of stac on Ca$_V$1.2 (*Campiglio et al., 2018*; *Polster et al., 2015*; *Wong King Yuen et al., 2017*). Devoid of stac, Ca$_V$1.2 exhibits CaM-mediated CDI manifesting as enhanced decay of Ca$^{2+}$ (red) versus Ba$^{2+}$ current (black) when elicited using a step depolarization (*Figure 1A*, middle subpanel). As Ba$^{2+}$ binds CaM poorly (*Linse and Forsén, 1995*), Ba$^{2+}$-currents furnish a baseline measure of voltage-dependent inactivation (VDI) without CDI. Upon stac2 coexpression, CDI is diminished (*Figure 1A*, right subpanel). To quantify steady-state extent of inactivation, we measured the fraction of peak Ca$^{2+}$ and Ba$^{2+}$ current remaining after 300 ms depolarization, $r_{Ca}$ and $r_{Ba}$ (*Figure 1—figure supplement 1A*). The strength of CDI is quantified as $CDI_{300} = 1 - r_{Ca}/r_{Ba}$, the fractional Ca$^{2+}$-dependent component of inactivation. Thus quantified, the population data confirm a reduction in CDI of Ca$_V$1.2 with stac2 (p=3.6 $\times$ 10$^{-5}$; *Figure 1B*). Further analysis shows that both stac1 and stac3 isoforms also diminish CDI (p=2.0 $\times$ 10$^{-5}$ and 7.1 $\times$ 10$^{-5}$, respectively, *Figure 1B* and *Figure 1—figure supplement 1A*). Similarly, Ca$_V$1.3 short variant (Ca$_V$1.3$_S$), a

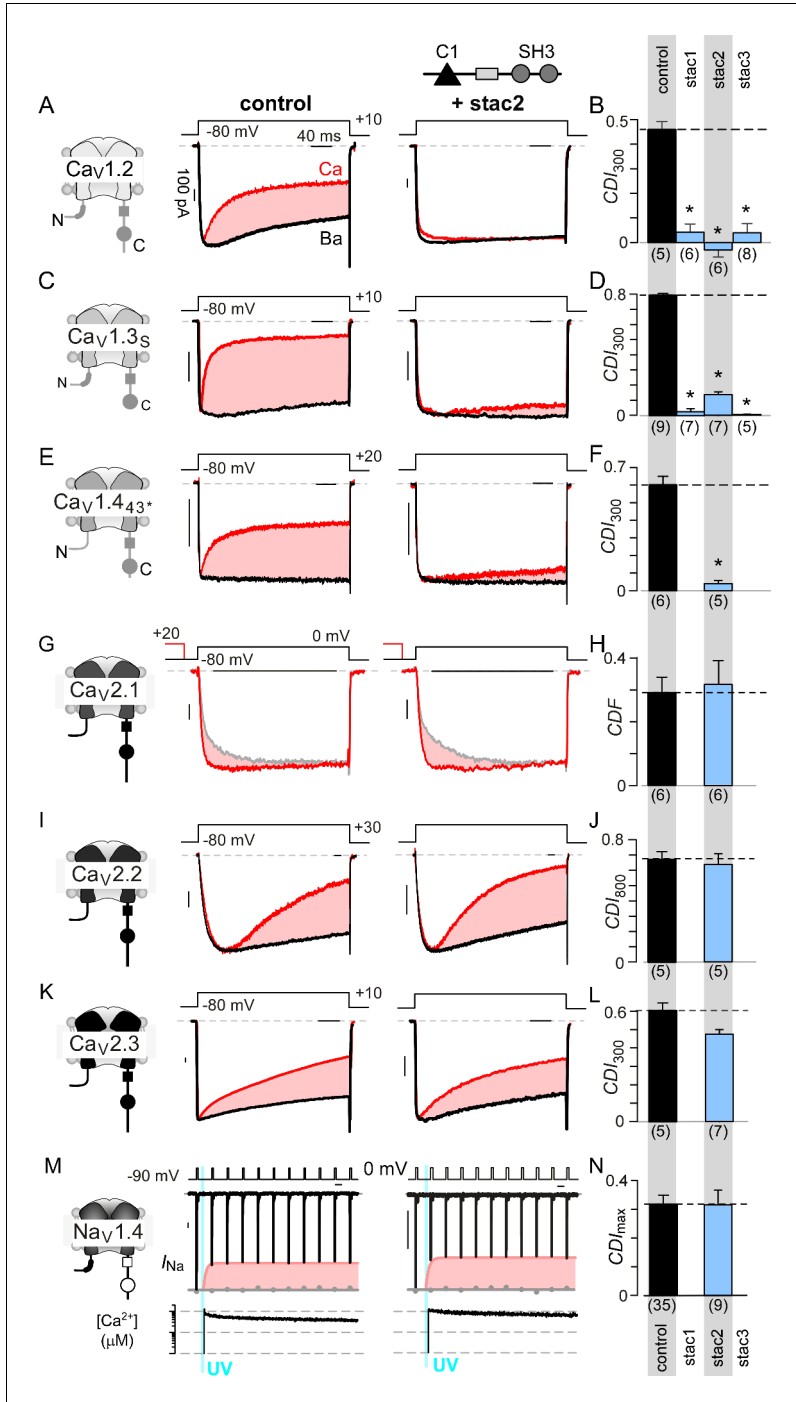

**Figure 1.** Stac specifically abolishes $Ca^{2+}$/CaM-regulation in $Ca_V1$ channels. (A) Stac2 diminishes CDI of $Ca_V1.2$. Left, cartoon schematic shows $Ca_V1.2$. Middle, exemplar current traces evoked in response to +10 mV voltage-step shows robust CDI (rose shaded area) evident as enhanced current decay with $Ca^{2+}$ (red) versus $Ba^{2+}$ (black) as the charge carrier. Right, stac2 abolishes CDI. Steady-state levels of inactivation are assessed as the fraction of peak current remaining following 300 ms depolarization ($r_{Ca}$ and $r_{Ba}$) and CDI = 1 – $r_{Ca}/r_{Ba}$. (B) Bar graph displays population data of $CDI_{300}$ for $Ca_V1.2$ in the absence and in the presence of stac1, stac2, or stac3. Dashed line shows baseline CDI in the absence of stac for comparison. Each bar, mean ±S.E.M. obtained from specified number of cells (n). (C–D) Stac isoforms suppress CDI of $Ca_V1.3_S$, the canonical short variant, as confirmed by both exemplar traces (C) and population data (D). Format as in (A) and (B). (E–F) Stac2 abolishes CDI of $Ca_V1.4_{43*}$ assessed in response to +20 mV test pulse. Format as in (A) and (B). (G–H) Stac2 spares CDF of $Ca_V2.1$, as evaluated using a prepulse protocol. An isolated test pulse to 0 mV elicits $Ca^{2+}$ currents with biphasic activation

*Figure 1 continued on next page*

*Figure 1 continued*

(gray, **G**). With a + 20 mV prepulse, channels are partially facilitated and the slow component of activation is reduced (red, **G**). The area between the two current traces ($\Delta Q$), divided by $\tau_{slow}$, yields facilitation (g). Bar graph plots, CDF = $g_{Ca} - g_{Ba}$**H**). Each bar, mean ±S.E.M from specified number of cells (n). (**I–J**) Stac2 spares CDI of Ca$_V$2.2 assessed in response to +30 mV test pulse. Here, CDI is evaluated following 800 ms of depolarization to accommodate slow inactivation kinetics, yielding CDI$_{800}$. Format as in (**A**) and (**B**). (**K–L**) Stac2 spares CDI of Ca$_V$2.3. Format as in (**A**) and (**B**). (**M–N**) Stac2 spares CDI of Na$_V$1.4. Both in the absence and presence of stac, Na$_V$1.4 peak currents decline following a Ca$^{2+}$ step (rose fit) (**M**). Gray dots, peak currents before uncaging. CDI = 1 – average peak $I_{Na}$ of last three to four responses after Ca$^{2+}$ uncaging / peak $I_{Na}$ before uncaging. Bar graph plots maximal CDI observed with Ca$^{2+}$ steps > 5 µM (**N**). Each bar, mean ±S.E.M.

DOI: https://doi.org/10.7554/eLife.35222.002

The following figure supplement is available for figure 1:

**Figure supplement 1.** Extended data highlight selectivity of stac in modulating Ca$_V$1 versus Ca$_V$2 and Na$_V$1 channels.

DOI: https://doi.org/10.7554/eLife.35222.003

close homolog of Ca$_V$1.2, exhibits strong baseline CDI that is reduced on co-expression of stac1, stac2, and stac3 (p<1 $\times$ 10$^{-5}$; *Figure 1C–D* and *Figure 1—figure supplement 1B*). Generalizing this phenomenon, stac2 also reduces CDI of Ca$_V$1.4$_{43*}$ (p=3.2 $\times$ 10$^{-5}$; *Figure 1E–F*; *Figure 1—figure supplement 1C*) (*Tan et al., 2012*).

Encouraged by its pervasiveness, we considered whether stac alters Ca$^{2+}$-dependent modulation of Ca$_V$2 isoforms that are abundant in the central nervous system. For Ca$_V$2.1, CaM elaborates both CDF and CDI (*DeMaria et al., 2001*; *Lee et al., 2000*). However, the Ca$^{2+}$-sensitivity of CDI process is over 50-fold weaker than that of CDF, casting this negative feedback beyond physiological bounds (*Lee et al., 2015*). As such, we probed whether stac tunes CDF of Ca$_V$2.1 using a well-established prepulse protocol (*DeMaria et al., 2001*; *Thomas and Lee, 2016*). *Figure 1G* displays wildtype Ca$_V$2.1 currents in the absence of stac2. On presentation of an isolated test pulse to 0 mV, the activation of Ca$^{2+}$ current follows a biphasic response (gray trace). Following a brief voltage prepulse, however, the ensuing test pulse yields enhanced Ca$^{2+}$-currents with monophasic activation reflecting CDF (red trace). Further quantification revealed no change in CDF of Ca$_V$2.1 following the addition of stac2 in both exemplar current recordings (*Figure 1G*) and population data (*Figure 1H*; *Figure 1—figure supplement 1D*). For Ca$_V$2.2, CaM-regulation manifests as a kinetically slow CDI (*Figure 1I*) (*Liang et al., 2003*), that persists despite stac co-expression (*Figure 1I–J*; *Figure 1—figure supplement 1E*). Here CDI is quantified by metric CDI$_{800}$ = 1 – $r_{Ca}/r_{Ba}$, measured following 800 ms of depolarization (*Figure 1Figure 1K–L*; *Figure 1—figure supplement 1F*).

Lastly, we tested whether stac suppresses Ca$^{2+}$-regulation of Na$_V$1, related to Ca$_V$1. Although all Na$_V$1 possess a CI module homologous to both Ca$_V$1 and Ca$_V$2 (*Babitch, 1990*), CDI that bears mechanistic similarity to Ca$_V$ has been identified only in Na$_V$1.4 (*Ben-Johny et al., 2014*). Unlike Ca$_V$, Na$_V$ channels do not convey Ca$^{2+}$ influx that triggers Ca$^{2+}$-feedback. We used rapid photo-uncaging of Ca$^{2+}$ to produce a step-like increase in intracellular [Ca$^{2+}$]$_i$, whose magnitude is simultaneously monitored via fluorescent indicators. *Figure 1M* displays baseline Ca$^{2+}$-regulation of Na$_V$1.4. As CDI is kinetically slow in comparison to fast inactivation, we applied a train of step depolarizations evoked at 10 Hz to probe Ca$^{2+}$-dependent effects (*Ben-Johny et al., 2014*). Without Ca$^{2+}$-uncaging, peak Na$_V$1.4 currents remained steady (gray dots). In response to an ~10 µM Ca$^{2+}$ step, the peak Na current declined rapidly revealing CDI (red envelope). Stac overexpression, however, does not disrupt Na$_V$1.4 CDI (*Figure 1M–N*; *Figure 1—figure supplement 1G*). Overall, these results show the specificity of stac in tuning Ca$^{2+}$-regulation of Ca$_V$1 channels.

## Stac interacts with Ca$_V$1 CI module to elicit CDI suppression

We sought to identify molecular mechanisms that underlie selective Ca$_V$1 modulation by stac. As the stac effect here is inferred based on overexpression analysis, we determined relative concentration requirements for stac binding to Ca$_V$1 holo-channel complexes within the milieu of living cells. We used live cell FRET 2-hybrid assay (*Erickson et al., 2001*) to probe the interaction of CFP-tagged stac3 with YFP-linked Ca$_V$1.3$_S$. As all three stac variants suppress the CDI of all Ca$_V$1 isoforms, we

chose $Ca_V1.3$ as YFP-tethered channels and a repertoire of YFP-tagged intracellular loop peptides are readily available for in-depth binding analysis (*Yang et al., 2014*). Stac3 was selected for its high potency in suppressing $Ca_V1.3$ CDI (*Figure 1D*). Accordingly, we quantified FRET efficiency ($E_D$) between FRET pairs co-expressed in individual cells. By leveraging stochastic expression of the FRET pairs in cells, we obtained a saturating Langmuir relationship between $E_D$ and the free acceptor concentration ($A_{free}$) permitting estimation of relative binding affinities ($K_{d,EFF}$). Thus probed, we obtained a $Ca_V1.3$ holo-channel affinity for stac3 of $K_{d,EFF}$ = 1458 ± 251 $D_{free}$ units proportional to ~47 nM (*Figure 2A*). By comparison, similar analysis of CaM binding to $Ca_V1.3$ showed $K_{d,EFF}$ = 700 $D_{free}$units ~ 22 nM (*Yang et al., 2014*). Consequently, stac's relatively high binding-affinity for $Ca_V1.3$ suggests that it may be a potent modulator even at low concentrations.

With holo-channel binding assured, we systematically scanned YFP-tagged $Ca_V1.3$ intracellular domains (*Yang et al., 2014*) for stac binding sites (*Figure 2B*; *Figure 2—figure supplement 1A*). We found that stac3 binds well to the CI region ($K_{d,EFF}$ = 20697 ± 3023 $D_{free}$~0.67 ± 0.1 µM, *Figure 2C*). By contrast, analysis of the amino-terminus, intracellular loops between domains I and II

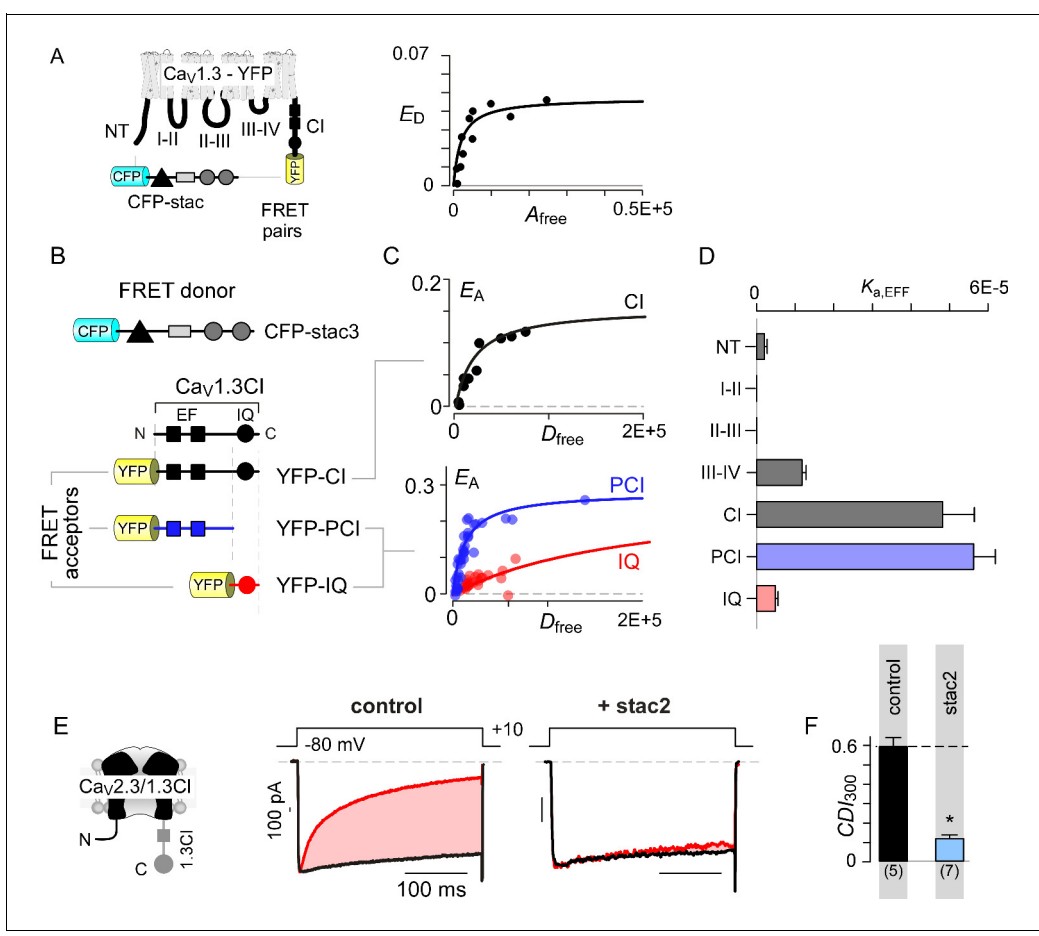

**Figure 2.** Stac interacts with the channel carboxy-terminus. (**A**) Live-cell FRET 2-hybrid assay shows high-affinity interaction between CFP-tagged stac3 with YFP-tethered holo-$Ca_V1.3$ channels in the presence of auxiliary $\beta_{2A}$ and $\alpha_2\delta$ subunits. (**B**) Cartoon shows FRET pairs, CFP-stac3 with YFP-CI, YFP-PCI, and YFP-IQ of $Ca_V1.3$. (**C**) FRET-binding curves show robust binding of stac3 to both the CI and PCI segment while binding to IQ is weaker. (**D**) Bar graph summarizes the relative association constant, $K_{a,EFF}$, of stac2 binding to major channel intracellular domains. (**E–F**) Transferring $Ca_V1.3_S$ CI to $Ca_V2.3$ ($Ca_V2.3/1.3$ CI) unveils latent stac2-mediated suppression of CDI. Format as in *Figure 1A – B*.

DOI: https://doi.org/10.7554/eLife.35222.004

The following figure supplement is available for figure 2:

**Figure supplement 1.** Systematic FRET 2-hybrid scan of major intracellular loops of $Ca_V1.3$ with stac.
DOI: https://doi.org/10.7554/eLife.35222.005

(I-II loop), domains II and III (II-III loop), and domains III and IV (III-IV loop) revealed far weaker binding (*Figure 2D*; *Figure 2—figure supplement 1B–E*). To further localize the putative binding loci, we subdivided the CI module into two: (1) a proximal CI segment (PCI) composed of dual vestigial EF hand and preIQ segments and (2) the IQ domain (IQ). The YFP-tagged PCI segment bound stac3 with approximately tenfold higher affinity ($K_{d,EFF}$ = 17725 ± 3990 $D_{free}$~0.58 ± 0.1 µM) than the downstream IQ domain ($K_{d,EFF}$ = 204739 ± 25465 $D_{free}$~6.67 ± 0.8 µM) (*Figure 2C–D*). In all, systematic FRET analysis reveals that stac binds to Ca$_V$1 CI relying on upstream elements including the dual vestigial EF hand and preIQ domains, an interface distinct from that for CaM (*Bazzazi et al., 2013*; *Minor and Findeisen, 2010*).

To test for functional relevance of stac binding to the Ca$_V$1 CI module, we sought to confer stac-sensitivity to a stac-insensitive channel via a chimeric approach. We turned to Ca$_V$2.3 that lacks strong stac-mediated CDI suppression, yet forms functional chimeras with Ca$_V$1 (*Mori et al., 2008*; *Yang et al., 2014*). We replaced the CI region of Ca$_V$2.3 with the corresponding segment from Ca$_V$1.3 (Ca$_V$2.3/1.3 CI). Devoid of stac, Ca$_V$2.3–1.3 CI channels exhibit CDI isolated by high intracellular buffering (*Figure 2E–F*; *Figure 2—figure supplement 1F*). In contrast to wildtype Ca$_V$2.3, stac2 co-expression attenuated CDI (p=4.7 × 10$^{-4}$, *Figure 2E–F*; *Figure 2—figure supplement 1F*), suggesting that Ca$_V$1 CI module is necessary for stac-mediated CDI suppression.

## Stac uses an allosteric mechanism to suppresses CaM signaling

Given that both CaM and stac share the CI module as an effector site, two disparate mechanistic possibilities may allow suppression of Ca$^{2+}$-regulation. First, stac may competitively displace Ca$^{2+}$-free CaM (apoCaM) from its preassociation site. Second, stac may supersede CaM signaling to the channel pore via an allosteric mechanism. Systematic FRET analysis suggests that stac preferentially binds upstream CI elements (*Figure 2D*) while high-affinity CaM preassociation is supported via the IQ domain (*Bazzazi et al., 2013*; *Minor and Findeisen, 2010*), hinting that the two modulatory proteins may bind concurrently. To rule out competitive displacement of CaM preassociation, we covalently tethered CaM onto the Ca$_V$1.3 carboxy-tail using a poly-glycine linker (Ca$_V$1.3$_S$-CaM) (*Mori et al., 2004*; *Yang et al., 2014*). This maneuver preserves CDI (*Figure 3A* left) and ensures a high local CaM concentration near Ca$_V$1 extending into the millimolar range, sufficient to protect the channel from a competitive inhibitor (*Mori et al., 2004*). Dominant-negative CaM$_{1234}$ with its Ca$^{2+}$-binding sites disabled, typically displaces intact apoCaM from the CI module thereby resulting in a loss of CDI for wildtype channels (*Figure 3—figure supplement 1A–B*) (*Yang et al., 2014*). CDI of Ca$_V$1.3$_S$-CaM persists despite CaM$_{1234}$ co-expression (gray bar, *Figure 3B*; *Figure 3—figure supplement 1C–D*). By contrast, CDI of Ca$_V$1.3$_S$-CaM is diminished by co-expression of stac2 (p=3.8 × 10$^{-6}$, *Figure 3A–B*; *Figure 3—figure supplement 1E*) and stac3 (p=4.5 × 10$^{-4}$, *Figure 3—figure supplement 1F*). As a further test, co-expression of untethered Ca$_V$1.3$_S$ with both CaM and stac2 also showed low CDI (*Figure 3—figure supplement 1G*). We observed a similar fate for Ca$_V$1.2-CaM with stac2 (p=1.5 × 10$^{-5}$, *Figure 3C–D*; *Figure 3—figure supplement 1H*). These findings suggest that stac does not need to dislodge CaM from its Ca$_V$1.3 carboxy-tail binding interface to exert functional modulation.

To test this possibility, we undertook FRET 2-hybrid assay comparing binding of CFP-tagged CaM to YFP-tagged Ca$_V$1.3 CI in the presence and absence of unlabeled stac3. If stac3 were to competitively dislodge CaM, then this binding is predicted to be weakened. At baseline, CaM binds to Ca$_V$1.3 CI with a relative dissociation constant, $K_{d,EFF}$ ~4000 ± 291 $D_{free}$ units (*Figure 3E*) (*Ben Johny et al., 2013*). Upon co-expression of CaM$_{1234}$, this baseline binding is weakened ~11 fold, yielding a relative affinity of 47153 ± 4815 $D_{free}$ units (*Figure 3F*). By contrast, co-expression of stac3 did not appreciably perturb CaM binding to the CI module with $K_{d,EFF}$ = 4182 ± 330 $D_{free}$ units (*Figure 3G*). These results suggest that both stac and CaM act concurrently via distinct sites on the channel carboxy-tail, in contradiction with a competitive mechanism.

## Elementary mechanisms underlying stac-regulation of Ca$_V$1

Beyond Ca$^{2+}$-dependent regulation, apoCaM binding tunes the baseline activity of Ca$_V$ channels (*Adams et al., 2014*). Absent stac, Ca$_V$1 lacking prebound CaM adopts a low $P_O$ configuration (empty configuration, $P_{O/E}$) while apoCaM binding switches channels into a high $P_O$ mode (CaM-bound configuration, $P_{O/A}$) (*Adams et al., 2014*). Ca$^{2+}$/CaM divests this initial enhancement in $P_O$

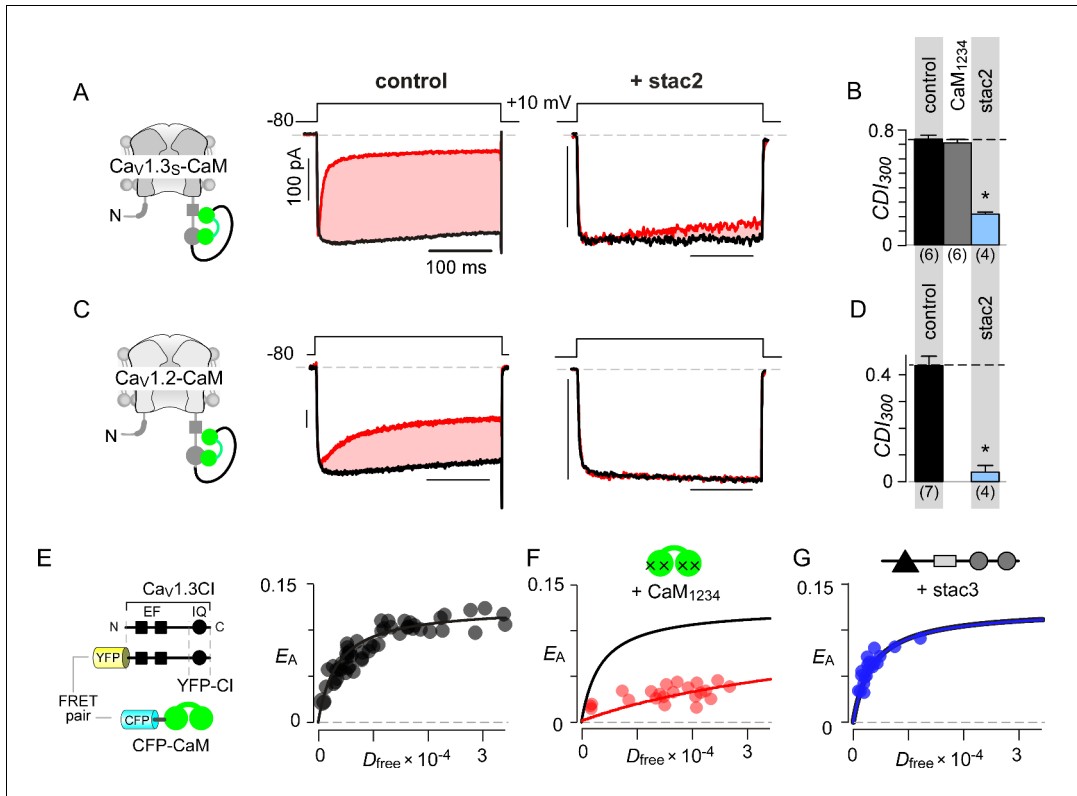

**Figure 3.** Allosteric regulation of stac by interaction with the channel carboxy-terminus. (**A–B**) Stac2 suppresses CDI of $Ca_V1.3_S$ tethered to CaM. In contrast, fusion of CaM protects $Ca_V1.3_S$ from competitive inhibitors such as $CaM_{1234}$. Format as in *Figure 1A–B*. (**C–D**) Stac2 suppresses CDI of $Ca_V1.2$ tethered to CaM. Format as in *Figure 1A–B*. (**E**) FRET 2-hybrid assay shows the high-affinity interaction of YFP-tagged $Ca_V1.3$ CI to CFP-tagged CaM with relative dissociation constant $K_{d,EFF}$ ~4000 ± 291 $D_{free}$ units. (**F**) Co-expression of untagged $CaM_{1234}$ with FRET pairs YFP-tagged $Ca_V1.3$ CI and CFP-tagged CaM results in a marked reduction in FRET efficiency. (**G**) Co-expression of untagged stac3 spares the binding of YFP-tagged $Ca_V1.3$ CI with CFP-tagged CaM, yielding an identical $E_A$-$D_{free}$ relationship to that in the absence of stac3 (**E**).
DOI: https://doi.org/10.7554/eLife.35222.006

The following figure supplement is available for figure 3:

**Figure supplement 1.** Extended data show that stac acts concurrently with CaM.
DOI: https://doi.org/10.7554/eLife.35222.007

and returns channels into a low $P_O$ gating mode ($P_{O/E}$) manifesting as CDI. The addition of stac as a regulatory agent enriches this modulatory scheme (*Figure 4A*). Three distinct scenarios may underlie suppression of $Ca^{2+}$-regulation by stac (*Figure 4B*): (1) stac binding may pre-inhibit channels into the low $P_O$ configuration ($P_{O/E}$) akin to $Ca^{2+}$-inactivated channels and prevent further $Ca^{2+}$-modulation, (2) stac may obstruct $Ca^{2+}$/CaM regulation while allowing apoCaM to change baseline $P_O$, (3) stac binding may allosterically lock channels into a high $P_O$ mode irrespective of CaM-binding status. For Scenario 3, it is possible that the baseline $P_O$ of $Ca_V1.3$ with stac may be distinct from that observed with CaM-overexpression. These three scenarios may be distinguished at the single-molecule level by assessing $Ca_V1$ $P_O$ under various CaM-bound conditions using low-noise electrophysiology. We focused on $Ca_V1.3$ given the rich assortment of post-transcriptionally modified variants with distinct CaM binding affinities (*Bazzazi et al., 2013*; *Liu et al., 2010*; *Singh et al., 2008*). We focused on three variants, $Ca_V1.3_S$ with high apoCaM affinity, and $Ca_V1.3_{MQDY}$ and $Ca_V1.3_L$ with low affinities. These variants possess distinct baseline $P_O$ and CDI and constitute a convenient platform to identify stac-dependent effects (*Adams et al., 2014*; *Tan et al., 2011*).

First, we analyzed $Ca_V1.3_S$ in the presence and absence of stac (*Figure 4C–E*) to determine whether stac may promote channel entry into a low $P_O$ gating configuration. $Ca_V1.3_S$ is typically pre-bound to CaM at endogenous CaM concentrations given its high affinity (*Adams et al., 2014*). $Ba^{2+}$

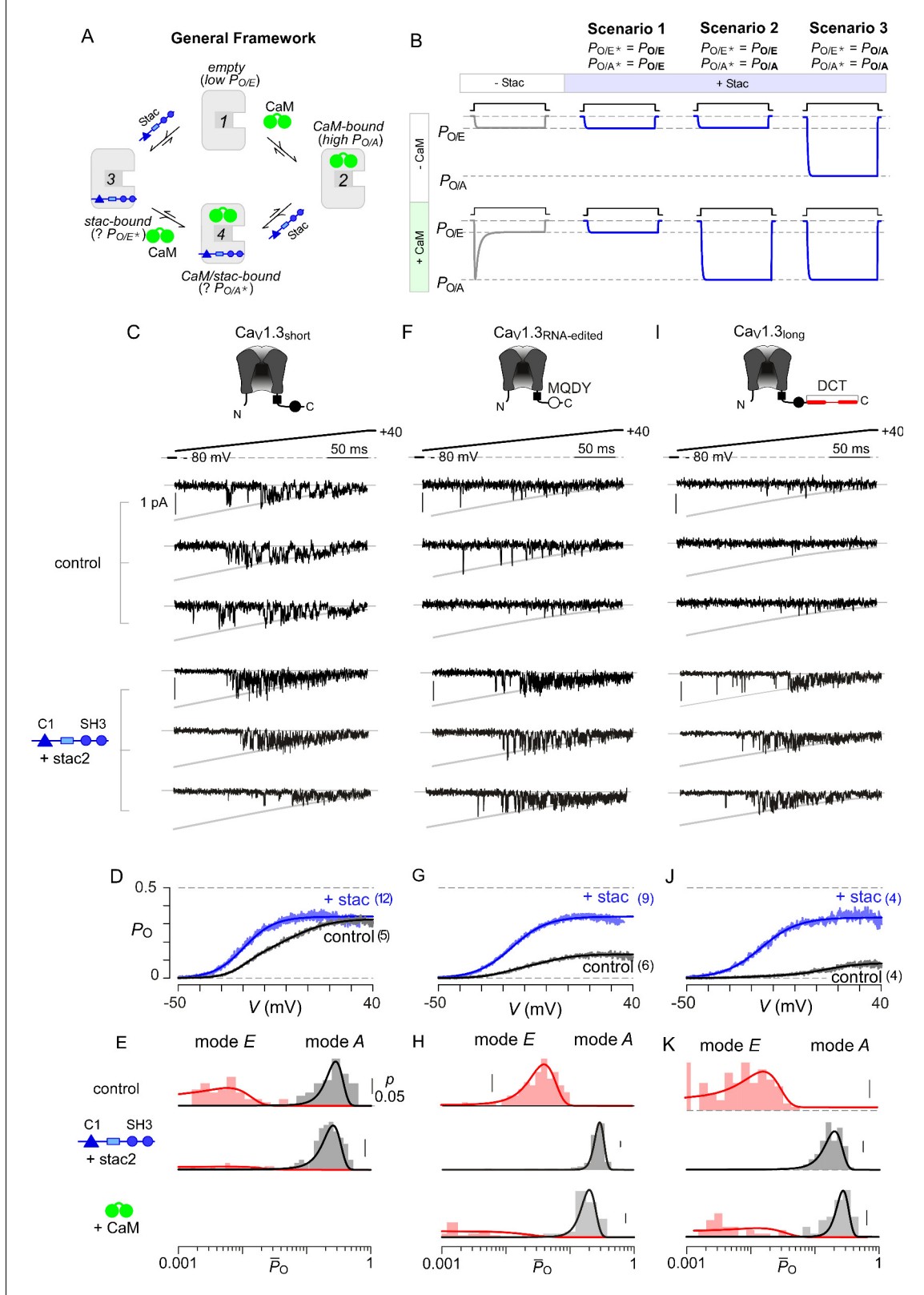

**Figure 4.** Stac enhances the $P_O$ of $Ca_V1.3$. (**A**) A general four-state scheme for stac and CaM modulation. (1) $Ca_V1.3_S$ devoid of CaM and stac possess a low baseline $P_O$ ($P_{O/E}$). (2) Without stac, apoCaM binding to $Ca_V1.3_S$ upregulates baseline $P_O$ ($P_{O/A}$). Baseline $P_O$ of $Ca_V1.3_S$ bound to stac in the absence (configuration 3, $P_{O/E*}$) and the presence of apoCaM (configuration 4, $P_{O/A*}$) are unknown. (**B**) Schematic outlines three mechanistic

*Figure 4 continued on next page*

*Figure 4 continued*

possibilities for stac binding to $Ca_V1$ and their functional outcomes. Scenario 1, stac uniformly suppresses $P_O$ of $Ca_V1$ ($P_{O/E}$) and abolishes CDI. Scenario 2, apoCaM tunes baseline $P_O$ of $Ca_V1$ despite concurrent stac binding. Stac, nonetheless, abrogates CDI. Scenario 3, stac uniformly upregulates the baseline $P_O$ of $Ca_V1$ and abolishes CDI ($P_{O/A}$). (C) Top, cartoon shows the canonical $Ca_V1.3_S$ variant with a high apoCaM binding affinity. Single-channel analysis of recombinant $Ca_V1.3_S$ in the absence (middle) and presence of stac (bottom). In both panels, the unitary $Ba^{2+}$ currents during voltage-ramp are shown between −50 mV and +40 mV (slanted gray lines, GHK fit). Robust $Ca_V1.3$ openings are detected in the absence and presence of stac. (D) Average single-channel $P_O$-voltage relationship for $Ca_V1.3_S$ obtained from multiple patches in the absence (gray) and presence of stac2 (blue). Error bars indicate ±S.E.M. for specified number of patches and 80–150 stochastic records per patch. (E) Histogram shows distribution of single-trial average $P_O$ ($\bar{P}_O$) for the voltage range -30 mV $\leq V \leq$ +25 mV under control (top), stac-bound (middle), and CaM-bound (bottom) conditions. $\bar{P}_O$-distribution is bimodal in the absence of stac corresponding to high $P_O$ (gray) and low $P_O$ (rose) gating modes. With stac, $\bar{P}_O$-distribution is largely restricted to the high $P_O$ mode. (F–H) Single-channel analysis of a recombinant $Ca_V1.3_{RNA-edited}$ variant reveals a marked upregulation in the baseline $P_O$ in the presence of stac compared with control conditions in which apoCaM preassociation is weak. Absent stac or CaM, single-trial $\bar{P}_O$-distribution is restricted to the low $P_O$ limits, With either stac or CaM, the high $P_O$ gating mode re-emerges. Format as in (C–E). (I–K) Stac also upregulates the baseline $P_O$ of $Ca_V1.3_L$, an alternatively spliced variant, by stabilizing the high $P_O$ gating configuration. Format as in (C–E).

DOI: https://doi.org/10.7554/eLife.35222.008

The following figure supplements are available for figure 4:

**Figure supplement 1.** Extended data show that stac2 preferentially biases a high $P_O$ gating mode for $Ca_V1.3$.

DOI: https://doi.org/10.7554/eLife.35222.009

**Figure supplement 2.** Extended data show that stac2 and CaM enhance the $P_O$ of $Ca_V1.3_{RNA-edited}$ variant via discreet transitions into a high $P_O$ gating mode.

DOI: https://doi.org/10.7554/eLife.35222.010

**Figure supplement 3.** Extended data show that both stac2 and CaM enhance the $P_O$ of $Ca_V1.3_L$ variant.

DOI: https://doi.org/10.7554/eLife.35222.011

is used as a charge carrier to estimate baseline behavior of channels without confounding effects of CDI. A slow voltage-ramp (shown between −50 and +40 mV) evokes stochastic channel openings that approximate near steady-state $P_O$ at each voltage. Stochastic records displayed in *Figure 4C* show channel openings as downward deflections to the open level (gray curves) and closures correspond to the zero-current portions of the trace. Robust openings are detected both in the presence and absence of stac (*Figure 4C*). To estimate the steady-state $P_O$ – voltage relationship, we averaged many stochastic records to obtain a mean current that is divided into the open level and averaged over multiple patches. $Ca_V1.3_S$ variant exhibits high $P_O$ in the absence of stac (*Figure 4D*) (*Adams et al., 2014*). Upon stac2 co-expression, the open probability remains high with ~10 mV hyperpolarizing shift in the voltage-dependence of activation (*Figure 4D*). We scrutinized single-channel trials to assess changes in gating modes. *Figure 4—figure supplement 1* displays 10 sequential trials of $Ca_V1.3$ single-channel activity evoked by voltage-ramps introduced at 12 s intervals both in the presence and absence of stac. In the absence of stac, $Ca_V1.3_S$ activity switches between epochs of high and low activity, as evident from the diary plot of average $P_O$ within individual trials ($\bar{P}_o$) (*Figure 4—figure supplement 1B*). Analysis of single-trial $\bar{P}_o$ distribution reveals a bimodal distribution denoting discrete high and low $P_O$ gating modes (*Figure 4E*). Upon stac overexpression, channel activity is high as evident from $\bar{P}_o$-diary plots (*Figure 4—figure supplement 1D*) and single-trial $\bar{P}_o$ distribution (*Figure 4E*). In contradiction with Scenario 1, stac-bound channels are not pre-inhibited; rather, channels preferentially adopt a high $P_O$ mode.

To distinguish between the second and third mechanistic possibilities, we considered $Ca_V1.3$ variants with weakened apoCaM binding affinity that largely reside in the low $P_O$ configuration (*Adams et al., 2014*). Accordingly, we tested the baseline $P_O$ of $Ca_V1.3_{MQDY}$, an RNA-edited variant whose central isoleucine within the IQ domain is substituted to a methionine, (*Bazzazi et al., 2013*; *Huang et al., 2012*) and an alternative splice variant $Ca_V1.3_L$ containing an autoinhibitory domain that competitively displaces CaM (*Liu et al., 2010*; *Singh et al., 2008*). In the absence of stac and under endogenous CaM levels, both $Ca_V1.3_{MQDY}$ (*Figure 4F–G*) and $Ca_V1.3_L$ (*Figure 4I–J*) open sparsely, exhibiting a diminished maximal $P_O$ consistent with channels lacking CaM (*Adams et al., 2014*; *Bock et al., 2011*). Indeed, single-channel trials of $Ca_V1.3_{MQDY}$ (*Figure 4—figure supplement 2A–C*) and $Ca_V1.3_L$ (*Figure 4—figure supplement 3A–C*) under endogenous levels of CaM reveal uniformly low activity, with single-trial $P_O$ distribution restricted to low limits (*Figure 4H* for $Ca_V1.3_{MQDY}$; *Figure 4K* for $Ca_V1.3_L$). CaM overexpression with both channel variants reveals the resurgence

of epochs of high activity (*Figure 4—figure supplement 2D–E*; *Figure 4—figure supplement 3D–E*) and a bimodal $P_O$ distribution with a substantial fraction of trials corresponding to a high $P_O$ configuration (*Figure 4H and K* for $Ca_V1.3_{MQDY}$ and $Ca_V1.3_L$ respectively). Upon stac co-expression, robust channel openings re-emerge for both $Ca_V1.3_{MQDY}$ (*Figure 4F–G*) and $Ca_V1.3_L$ (*Figure 4I–J*) yielding an enhanced baseline $P_O$ akin to $Ca_V1.3_S$ variant (*Adams et al., 2014*). Scrutiny of single-channel trials for both channel variants reveal uniformly high channel activity (*Figure 4—figure supplement 2F–G* for $Ca_V1.3_{MQDY}$; *Figure 4—figure supplement 3F–G* for $Ca_V1.3_L$) and single-trial $P_O$ distributions are now within the high activity limits (*Figure 4H and K*) reminiscent of the CaM overexpression. The steady-state $P_O$-$V$ relations for $Ca_V1.3_S$, $Ca_V1.3_{MQDY}$, and $Ca_V1.3_L$ in the presence of stac closely approximate each other (*Figures 4D, G and J*). These findings demonstrate that consistent with Scenario 3, stac-binding locks $Ca_V1.3$ channels in the high $P_O$ configuration and effectively decouples the channel pore from CaM-dependent conformational changes. Moreover, these results highlight the dominance of stac over CaM in $Ca_V1$ modulation.

## U-domain constitutes a minimal motif for $Ca_V1$ CDI suppression

With elementary mechanisms discerned, we turned to identify stac motifs functionally critical for allosteric suppression of CaM-regulation. Structurally, stac isoforms share a modular architecture composed of a C1 domain linked to two SH3 domains via a largely unstructured linker segment (U-linker region) (*Suzuki et al., 1996*). As these modular subcomponents typically recognize distinct ligands, we reasoned that their molecular functions may be separable (*Cohen et al., 1995*; *Colon-Gonzalez and Kazanietz, 2006*). We trisected stac2 to assess whether individual subcomponents can recapitulate functional regulation. We focused initially on C1 and tandem SH3 domains as these segments were recently shown to be critical for $Ca_V1.1$ binding and triadic localization in skeletal muscle (*Campiglio and Flucher, 2017*; *Wong King Yuen et al., 2017*). Co-expression of either segment, however, only minimally perturbed CDI of $Ca_V1.2$-CaM (*Figures 5A, C and D*; *Figure 5—figure supplement 1A–1C*). By contrast, the linker region by itself fully abolished CDI of these channels (p=$8.9 \times 10^{-6}$, *Figure 5B and D*; *Figure 5—figure supplement 1D*), recapitulating the effect of stac2 on $Ca_V1.2$.

To localize functional segments within the U-linker, we undertook bioinformatic analysis to identify highly conserved regions. We performed multiple sequence alignment of complete sequences of 770 stac orthologs using the MUSCLE algorithm (*Edgar, 2004*) and subsequently computed an empirical measure for the degree of protein sequence conservation at each position. The degree of conservation is defined as the likelihood of observing the consensus residue at each sequence position divided by the number of distinct residues observed at this position. By this algorithm, perfectly conserved residues will yield a unitary value, whereas poorly conserved residues will have a lower score. We identified a 22-amino acid stretch, termed the U-domain ('unknown' domain), exhibiting high conservation (*Figure 5E*, blue shaded region). Co-expression of U-domain diminishes CDI of both $Ca_V1.2$-CaM and $Ca_V1.3$-CaM (*Figure 5F–H*, *Figure 5—figure supplement 1E–G*). Thus informed, we undertook systematic alanine scanning mutagenesis of the stac2 U-domain to identify key residues (*Figure 5I*; *Figure 5—figure supplement 2*). Co-expression of mutant stac2 with triple alanine substitution of residues ETL[206-208] resulted in minimal disruption of $Ca_V1.2$ and $Ca_V1.3$ CDI (*Figure 5J–K*), suggesting that these residues are necessary. Further analysis revealed residues PVY[203-205] and KVD[200-202] to be critical for stac function (*Figure 5I*; *Figure 5—figure supplement 2A–2C*). Residues outside these loci minimally affected stac modulation of $Ca_V1$ (*Figure 5I*; *Figure 5—figure supplement 2D–G*). These findings confirm the necessity and sufficiency of U-domain as a minimal motif for preventing CaM-regulation of $Ca_V1$.

## U-domain modulates native $Ca_V1$ and reshapes cardiac action potentials

Having identified a minimal U-domain for CDI suppression, we sought to assess potential physiological consequences of stac regulation in cardiac myocytes. As stac expression is yet to be identified in myocytes, we first probed its presence using immunohistochemistry with stac1- and stac2-specific antibodies. To ensure that the two antibodies reliably probe the two isoforms, we first evaluated the ability to detect stac isoforms exogenously expressed in HEK293 cells (*Figure 6—figure supplement 1*). Untransfected cells show minimal stac1 and stac2 immunostaining (*Figure 6—figure supplement 1A–B*), as confirmed by confocal imaging and population data. By contrast, immunostaining with

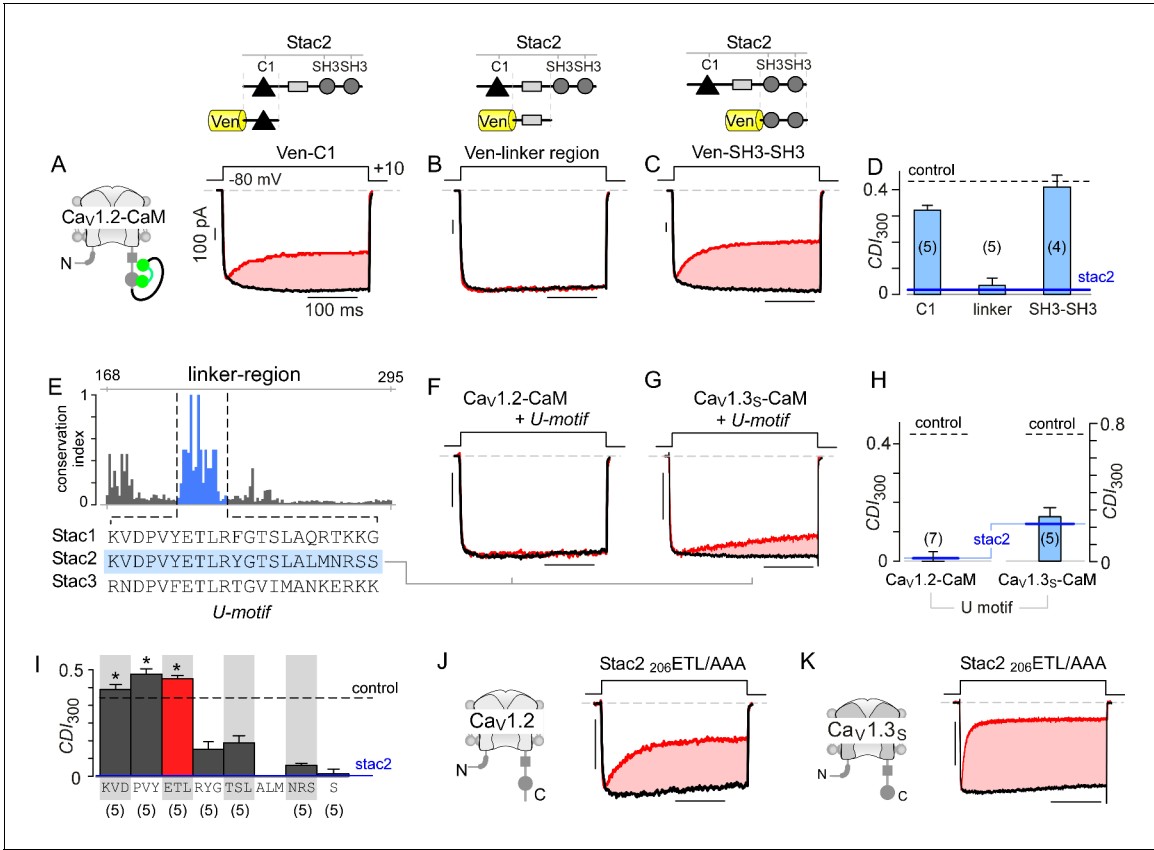

**Figure 5.** Stac U-domain is a minimal effect domain for suppression of Ca$_V$1 CDI. (**A–C**) To localize an effector motif for stac2, CDI of Ca$_V$1.2-CaM was quantified in the presence of three stac subdomains: (1) C1, (2) linker region, and (3) SH3-SH3. Exemplar traces in response to a +10 mV voltage-step depolarization show robust CDI of Ca$_V$1.2-CaM in the presence of C1 (**A**), and SH3-SH3 (**C**) domains. Co-expression of the linker-region is sufficient to suppress CDI of Ca$_V$1.2-CaM (**B**). Format as in *Figure 1A*. (**D**) Bar graph summarizes population data for Ca$_V$1.2-CaM CDI in the presence of the three stac subdomains. Each bar, mean ±S.E.M of CDI$_{300}$ at +10 mV from specified number of cells. CDI levels in the presence (solid blue line) and absence (dashed gray line) of full-length stac2 is reproduced for comparison. (**E**) Bar graph shows degree of conservation for the linker region across 770 orthologs of stac2. A well conserved subsegment termed U-domain is shaded blue. (**F–G**) Co-expression of U-domain is sufficient to abolish CDI of Ca$_V$1.2-CaM (**F**) and Ca$_V$1.3-CaM (**G**). Format as in *Figure 1A*. (**H**) Bar graph displays population data for CDI of Ca$_V$1.2-CaM and Ca$_V$1.3-CaM in the presence of U-domain. Each bar, mean ±S.E.M of CDI$_{300}$ at +10 mV from specified number of cells. Dashed line, baseline CDI for both channels in the absence of stac2. Blue line, CDI of both channels in the presence of full-length stac2. (**I**) Systematic alanine scanning mutagenesis of the U-domain reveals critical determinants for stac-mediated suppression of Ca$_V$1.2 CDI. For comparison, Ca$_V$1.2 CDI in the presence (blue line) and absence (black dashed line) of stac2 are shown. Stac2 mutants $_{200}$KVD/AAA, $_{203}$PVY/AAA, $_{206}$ETL/AAA fully abolish stac2-mediated CDI suppression. (**J**) Exemplar currents show that stac2 mutant $_{206}$ETL/AAA eliminates stac's ability to suppress Ca$_V$1.2 CDI. Format as in *Figure 1A*. (**K**) Stac2 $_{206}$ETL/AAA also fails to inhibit CDI of Ca$_V$1.3$_S$. Format as in *Figure 1A*.

DOI: https://doi.org/10.7554/eLife.35222.012

The following figure supplements are available for figure 5:

**Figure supplement 1.** Extended data demonstrate that the U-motif is a minimal domain for suppressing CDI of Ca$_V$1.2 and Ca$_V$1.3.
DOI: https://doi.org/10.7554/eLife.35222.013
**Figure supplement 2.** Systematic alanine scanning mutagenesis of minimal U-motif reveals structural determinants for stac modulation of Ca$_V$1.
DOI: https://doi.org/10.7554/eLife.35222.014

stac1 antibody shows labeling with cells expressing stac1 but not stac2 or stac3. Similarly, labeling with stac2 antibody reveals substantial fluorescence (*F* > 300) in cells transfected with stac2 but not stac1 or stac3. Thus informed, we assessed expression and localization of stac isoforms in cardiac myocytes (*Figure 6—figure supplement 1C–F*). Analysis of acutely dissociated adult guinea pig ventricular myocytes (aGPVMs) showed stac2 labeling but not stac1 (*Figure 6—figure supplement 1C*). Consistent with these findings, immunoblotting with stac2 antibody showed ~50 kDa signal in stac2-transfected HEK293 cells but absent from untransfected cells, confirming the ability of the antibody to detect stac2 (*Figure 6—figure supplement 1G*). Analysis of aGPVM lysate revealed likely

endogenous stac2 with a similar molecular weight to that of recombinant stac2 in HEK293 cells (*Figure 6—figure supplement 1G*).

Given this baseline expression, we next considered potential effects of fluctuations in ambient stac levels. We synthesized the U-domain of stac2 as a peptide and delivered it via pipette dialysis to acutely elevate the myocyte's cytosolic concentration. We validated the synthesized peptide by testing its effect on recombinant Ca$_V$1.2 expressed in HEK293 cells (*Figure 6A*). Following pipette dialysis of the U-peptide, CDI of Ca$_V$1.2 was reduced as evident from exemplar currents and population data (*Figure 6B–C*; *Figure 6—figure supplement 2A–B*). Thus affirmed, we isolated ventricular

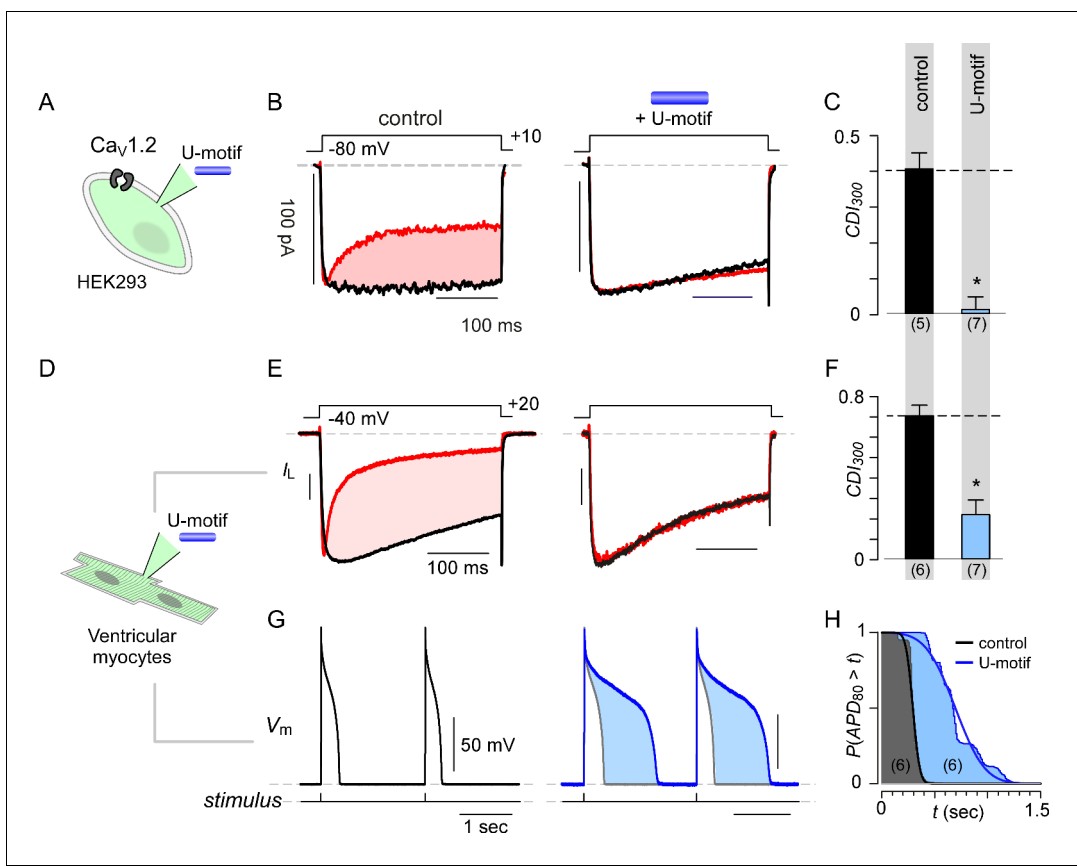

**Figure 6.** Synthetic U-domain peptide is sufficient for physiological perturbations. (**A**) Schematic illustrates pipette dialysis of custom synthesized U-domain peptide in Ca$_V$1.2 heterologously expressed in HEK293 cells, a strategy that emulates acute elevation of cytosolic stac2 levels. (**B–C**) Exemplar traces and population data confirm that pipette dialysis of U-domain suppresses CDI of recombinant Ca$_V$1.2 in HEK293 cells. Format as in *Figure 1A–B*. Control relation in (**C**) is duplicated from *Figure 1B*. (**D–F**) Pipette dialysis of U-domain abolishes CDI of endogenous L-type current in freshly dissociated ventricular myocytes from adult guinea pigs as evident from exemplar traces and bar graph summary of population data. To eliminate T-type current, the cells were depolarized to −40 mV for a period of 100 ms. Format as in (**A–C**). (**G**) Exemplar action potential traces of aGPVMs paced at 0.5 Hz with (blue) and without (black) 0.5 μM U-domain in the internal solution. In the presence of U-domain, the action potentials are markedly prolonged (blue shaded area) consistent with a loss of CDI of native L-type current. (**H**) Complement of cumulative distribution ($P(APD_{80} > t)$) of action potential durations ($APD_{80}$) obtained in the presence (blue) and absence (black) of U-domain in the internal solution.
DOI: https://doi.org/10.7554/eLife.35222.015

The following figure supplements are available for figure 6:

**Figure supplement 1.** Baseline expression of stac in cardiac myocytes.
DOI: https://doi.org/10.7554/eLife.35222.016
**Figure supplement 2.** Pipette dialysis of U-motif as a peptide abolishes CDI of both recombinant and native L-type currents in ventricular myocytes.
DOI: https://doi.org/10.7554/eLife.35222.017

myocytes from adult guinea pigs (aGPVMs) to probe changes in CDI of native $Ca_V$ channels and action potential duration in response to changes in stac levels (*Figure 6D*). Devoid of U-domain peptide, endogenous $Ca^{2+}$ currents in ventricular myocytes displayed CDI, establishing baseline levels of CaM-regulation (*Figure 6E*, *Figure 6—figure supplement 2D*). Pipette dialysis of U-peptide reduced CDI in myocytes (*Figure 6E–F*, *Figure 6—figure supplement 2E*). The reduction in overall inactivation of $Ca^{2+}$ currents suggest that fluctuations in stac levels may markedly alter action potential waveforms. To test this possibility, we obtained current-clamp recordings of aGPVMs and compared action potential waveforms in the presence and absence of U-peptide. *Figure 6G* shows typical voltage profiles of action potentials in aGPVMs paced at 0.5 Hz. Waveforms are stable between traces and the mean action potential duration ($APD_{80}$), the duration of time when the action potential is at least 80% of its peak voltage, is $277.9 \pm 31.37$ ms (mean $\pm$S.E.M., n = 6). *Figure 6H* displays the complement of the cumulative distribution of $APD_{80}$. When the peptide is added to the internal solution, $APD_{80}$ is enhanced to $740.1 \pm 105.49$ ms (n = 6) (*Figure 6G–H*). Thus, the U-peptide both alters the CDI of endogenous cardiac $Ca_V1$, prolongs APD, and may ultimately destabilize rhythmicity of the heart.

## Fhf selectively abrogate CaM signaling to $Na_V1$

Encouraged by the selectivity of stac for $Ca_V1$, we sought to identify other regulatory proteins that may tune CaM-signaling to related channel families. However, recognizing such modulators amidst ion channel signalosomes is challenging. Given that stac interacts with $Ca_V1$ CI module via the PCI element, we reasoned that other $Ca_V$ and $Na_V$ interacting proteins that engage a similar interface may suppress CaM-feedback. Intriguingly, recent atomic structures show that fhf interacts with $Na_V1$ CI module via the PCI interface (*Figure 7A*) (*Wang et al., 2012*). Yet, functionally, fhf isoforms are thought to modulate only voltage-dependent gating properties, with effects on $Ca^{2+}$/CaM-regulation unknown (*Goldfarb et al., 2007*; *Lou et al., 2005*; *Wang et al., 2012*). To test whether fhf alters $Na_V$ CDI, we undertook quantitative $Ca^{2+}$ photo-uncaging of the skeletal muscle $Na_V1.4$ isoform. We focused here on fhf1b given its modest baseline expression in skeletal muscle and pathological enrichment in critical illness myopathies (*Kraner et al., 2012*). *Figure 7B* reproduces baseline levels of CDI for $Na_V1.4$ under control conditions. Co-expression of fhf1b abolished CDI (*Figure 7B–C*; *Figure 7—figure supplement 1A*), unveiling a novel role of fhf in tuning $Ca^{2+}$-feedback of $Na_V$ channels. To assess selectivity, we probed whether fhf alters CDI of $Ca_V1.3$. In comparison to control conditions, fhf co-expression spared $Ca_V1.3$ CDI (*Figure 7D–E*; *Figure 7—figure supplement 1B*) suggesting that fhf may be a selective modulator of $Na_V1$.

Mechanistically, functional results along with atomic structures of $Na_V1$ CI bound to CaM and fhf yield insights on mechanisms for CDI suppression (*Gabelli et al., 2014*; *Wang et al., 2012*; *Wang et al., 2014*). Both fhf and CaM bind concurrently to $Na_V1$ CI (*Wang et al., 2012*; *Wang et al., 2014*), with fhf binding triggering a conformational rearrangement of CaM (*Figure 7A*) (*Gabelli et al., 2014*; *Wang et al., 2012*). To experimentally validate allostery, we followed our strategy with $Ca_V1.3$ and tethered CaM to $Na_V1.4$ carboxy-tail. Reassuringly $Na_V1.4$-CaM exhibits robust baseline CDI (*Figure 7F*; *Figure 7—figure supplement 1C*). Whereas dominant negative $CaM_{1234}$ typically abolishes CDI of $Na_V1.4$ (*Ben-Johny et al., 2014*), $Na_V1.4$-CaM exhibits robust CDI despite $CaM_{1234}$, confirming the protective nature of tethered CaM against competitive inhibitors (*Figure 7G*; *Figure 7—figure supplement 1D*). Co-expression of fhf1b, however, reduces CDI of $Na_V1.4$-CaM (*Figure 7F–G*; *Figure 7—figure supplement 1E*). Thus, like stac modulation of $Ca_V1$, fhf overrides CaM signaling to $Na_V1.4$ despite a tethered CaM, suggesting that fhf acts in allostery.

To garner a structural perspective, we turn to $Na_V1.5$ CI/fhf complex (*Figure 7A*) as the atomistic basis of the stac/$Ca_V1$ CI interaction is unknown (*Wang et al., 2012*; *Wang et al., 2014*; *Wong King Yuen et al., 2017*). Whereas the dual-vestigial EF hand segments of $Na_V1.5$ and $Ca_V1.1$ are similar (*Figure 7H–I*), the fhf binding interface of $Na_V1.5$, including the preIQ loop diverges from corresponding segments of $Ca_V1.1$ and introduces a steric clash (*Figure 7I–J*) (*Wang et al., 2012*; *Wu et al., 2016*). Thus, by leveraging structurally distinct loci on the CI module, fhf selectively diminish CaM signaling to $Na_V$ channels. These findings point to a class of auxiliary proteins that selectively adjust $Ca^{2+}$-dependent feedback to individual ion channel targets.

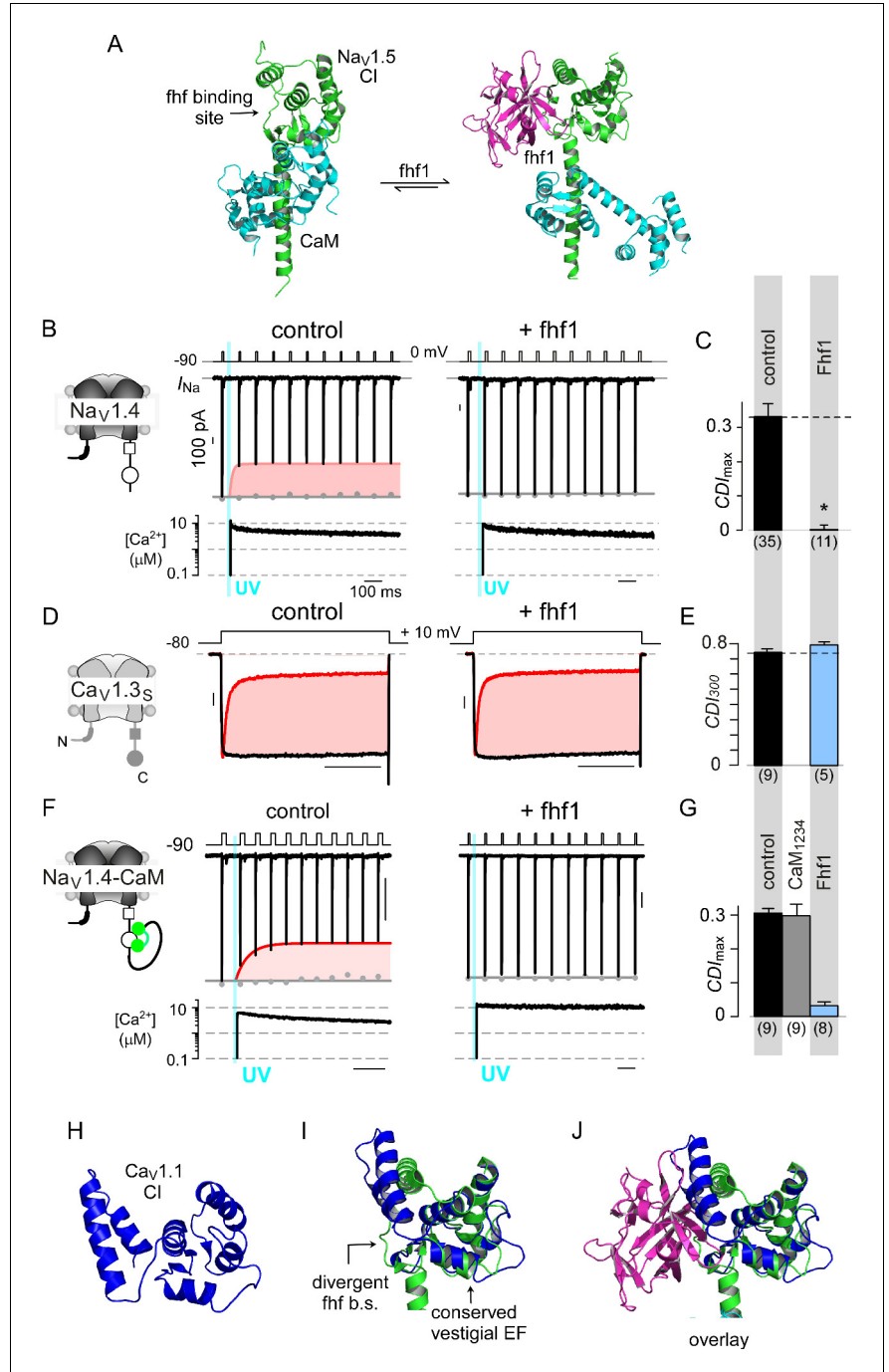

**Figure 7.** Fhf uses an allosteric mechanism to abrogate $Ca^{2+}$-feedback of $Na_V1.4$. (**A**) Structural comparison of $Na_V1.5$ CI (green) in the presence of CaM alone (cyan, left) or both CaM (cyan) and fhf1b (purple). Fhf binding changes baseline conformation of CaM on $Na_V1$ CI. (**B–C**) Co-expression of fhf1b abolishes CDI in $Na_V1.4$ evoked via $Ca^{2+}$ photo-uncaging. Format as in *Figure 1M–N*. Control data are reproduced from *Figure 1M–N* for comparison. (**D–E**) In sharp contrast, strong overexpression of fhf1b does not alter CDI of $Ca_V1.3_S$. Format as in *Figure 1A–B*. Control data reproduced from *Figure 1D* for comparison. (**F–G**) Fhf1 suppresses CDI of $Na_V1.4$ tethered to CaM. Fusion of CaM protects $Na_V1.4$ from competitive inhibitors such as $CaM_{1234}$ (**G**). Format as in *Figure 1M–N*. (**H**) Structure of $Ca_V1.1$ upstream CI elements (blue) composed of dual vestigial EF hands and preIQ segments isolated from cryo-EM structure of $Ca_V1.1$ (PDBID, 5GJV). This domain is the primary interface for stac interaction in the $Ca_V1$ CI. (**I**) Structural overlay of upstream CI elements of $Ca_V1.1$ (PDBID, 5GJV) and $Na_V1.5$ (PDBID, 4DCK) shows highly conserved dual vestigial EF hand segments while the fhf binding site is structurally

*Figure 7 continued on next page*

*Figure 7 continued*

divergent. (J) The divergence in the fhf binding interface in Ca$_V$1.1 in comparison to Na$_V$1.5 would introduce a steric clash that prohibits fhf binding to Ca$_V$ channels.

DOI: https://doi.org/10.7554/eLife.35222.018

The following figure supplement is available for figure 7:

**Figure supplement 1.** Extended data confirm the selectivity of fhf in modulating Na$_V$ versus Ca$_V$ channels.

DOI: https://doi.org/10.7554/eLife.35222.019

## Engineering synthetic modulation of Ca$_V$ channels

As both stac and fhf tune Ca$^{2+}$-feedback to individual Ca$_V$ and Na$_V$ targets by interacting with respective PCI segments, this mechanism furnishes a strategy to engineer synthetic channel modulators. We reasoned that introducing a short interaction motif into the PCI locus may permit inhibition of Ca$_V$1 Ca$^{2+}$-feedback by a novel protein. We chose the well-characterized RxxK motif from SLP-76 for its small size and high-affinity interaction with SH3 domain of Mona (*Harkiolaki et al., 2003*) (*Figure 8A*). Co-expression of Mona SH3 with wildtype Ca$_V$1.3$_S$ demonstrated the persistence of CDI, confirming the suitability of these channels as a 'blank slate' to confer synthetic modulation (*Figure 8B–C*; *Figure 8—figure supplement 1A*). We replaced a 12-residue segment in the preIQ domain with the RxxK motif, as highlighted in *Figure 8A*, yielding Ca$_V$1.3$_{RxxK}$ engineered channels. As this locus is situated upstream of the IQ domain, this maneuver spares apoCaM prebinding. Under endogenous levels of CaM, Ca$_V$1.3$_{RxxK}$ exhibit robust baseline CDI (*Figure 8D–E*; *Figure 8—figure supplement 1B*). Co-expression of Mona SH3 with Ca$_V$1.3$_{RxxK}$ markedly diminished CDI (*Figure 8D–E*; *Figure 8—figure supplement 1B*) , thus revealing engineered CDI suppression. These findings illustrate the versatility of the CI module as a regulatory hub and highlight the feasibility of developing synthetic modulators to tune Ca$^{2+}$-feedback of ion channels.

## Discussion

CaM is a dynamic regulator of Ca$_V$1, Ca$_V$2, and Na$_V$1, affording millisecond-precision Ca$^{2+}$-feedback of channel activity. Our findings suggest that distinct auxiliary regulatory proteins tune CaM signaling to individual targets selectively. Stac prevents CaM signaling to Ca$_V$1, while fhf reduces signaling to Na$_V$1 (*Figure 8F*). Parallel analysis of the two proteins delineates mechanisms and sets the stage for in-depth physiological analysis.

### Relationship to prior studies of stac-Ca$_V$ modulation

Stac regulation of Ca$_V$1 modifies multiple aspects of Ca$_V$1 function. For Ca$_V$1.1, stac3 enhances plasmalemmal trafficking (*Linsley et al., 2017b*; *Niu et al., 2018*; *Polster et al., 2015*; *Wong King Yuen et al., 2017*; *Wu et al., 2018*), and promotes conformational coupling to RyR (*Linsley et al., 2017a*; *Polster et al., 2016*). For Ca$_V$1.2, however, stac1-3 isoforms slow inactivation (*Campiglio et al., 2018*; *Polster et al., 2015*; *Wong King Yuen et al., 2017*). Our work generalizes the latter effect to the Ca$_V$1 family and further identifies a change in baseline channel openings ($P_O$).

A few mechanistic nuances merit attention. First, stac binds to multiple Ca$_V$1 segments including (1) the II-III linker (*Polster et al., 2018*; *Wong King Yuen et al., 2017*), (2) the III-IV linker (*Figure 2D*), and (3) the carboxy-tail (*Figure 2D*) (*Campiglio et al., 2018*; *Niu et al., 2018*). Previous studies have shown that stac interaction with the II-III linker is important for Ca$_V$1 trafficking in skeletal muscle (*Polster et al., 2018*; *Wong King Yuen et al., 2017*). Chimeric analysis here suggests that stac interaction with the carboxy-tail is critical for tuning CDI. Prior analysis of Ca$_V$1.2 triadic localization in myotubes suggested that the channel IQ domain may be important for stac binding (*Campiglio et al., 2018*). However, FRET 2-hybrid assay indicates that stac interaction with the IQ is around tenfold weaker than with the PCI segment. Second, prior work also suggested that stac-mediated reduction in CDI results from competitive displacement of CaM by stac (*Campiglio et al., 2018*). Functional experiments using Ca$_V$1 tethered to CaM, however, suggest that stac does not compete with CaM. Consistent with this scheme, FRET 2-hybrid analysis shows that CaM binding with the CI module is intact even in the presence of stac. Third, key domains within stac relevant for Ca$_V$ modulation remain controversial. Previous studies have identified the dual SH3 and C1 domains

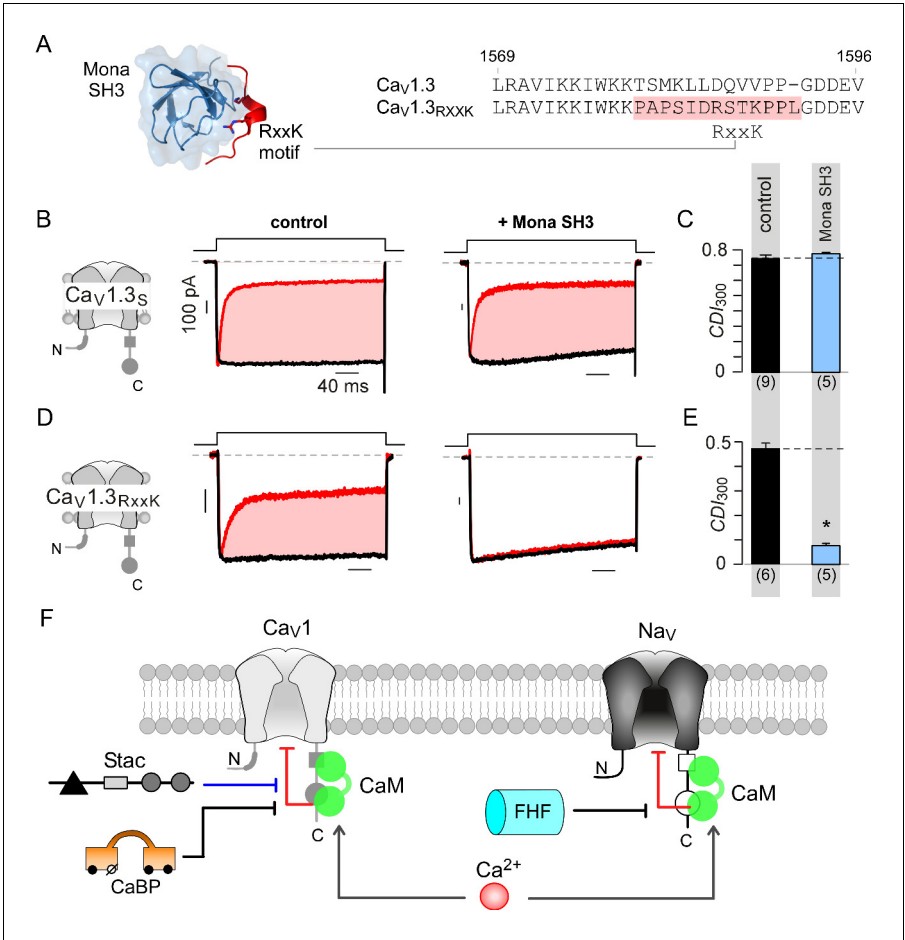

**Figure 8.** Engineering synthetic modulation of Ca$_V$1 channels. (**A**) Left, schematic shows the atomic structure of Mona SH3 domain in complex with RxxK motif. Right, sequence alignment outlines strategy for insertion of RxxK motif into Ca$_V$1.3, yielding Ca$_V$1.3$_{RxxK}$ to confer synthetic suppression of Ca$_V$1.3 CDI by Mona SH3. (**B–C**) Ca$_V$1.3$_S$ expressed with and without Mona SH3 shows full CDI, confirming that wildtype Ca$_V$1.3 CDI is insensitive to Mona SH3. Format as in *Figure 1A–B*. Control data are reproduced from *Figure 1D* for comparison. (**D–E**) Mona SH3 strongly diminishes CDI of Ca$_V$1.3$_{RxxK}$. Format as in *Figure 1A–B*. (**F**) Cartoon summarizes selective modulation of Ca$^{2+}$/CaM signaling to Ca$_V$1, and Na$_V$1 channels with CaM, stac, and fhf.

DOI: https://doi.org/10.7554/eLife.35222.020

The following figure supplement is available for figure 8:

**Figure supplement 1.** Extended data demonstrate feasibility of engineering synthetic modulators of CaM signaling to Ca$_V$1.3.

DOI: https://doi.org/10.7554/eLife.35222.021

to be critical for stac effect on trafficking and coupling to RyR (*Campiglio and Flucher, 2017*; *Linsley et al., 2017a*; *Linsley et al., 2017b*; *Polster et al., 2016*), while the C1 has been proposed to be critical for modifying Ca$_V$1 CDI (*Campiglio et al., 2018*; *Wong King Yuen et al., 2017*). Our findings instead suggest that the U-domain in the stac2 linker region is sufficient to fully recapitulate reduction in Ca$_V$1 CDI. Notably, prior analysis of the C1 domain also included this linker (*Wong King Yuen et al., 2017*). Given these experimental findings, a simple possibility is that distinct subdomains within stac interact with disparate channel segments to support multifunctionality of stac. While the U-domain modifies channel inactivation, other subdomains may support plasmalemmal trafficking and conformational coupling.

## Defining a general class of auxiliary modulators of CaM signaling

Although functionally divergent, $Ca_V1$, $Ca_V2$, and $Na_V1$ feature a modular CI element with a common CaM interaction fingerprint and subsequently, shared mechanistic basis for $Ca^{2+}$-regulation. For all three families, apoCaM prebinds the CI module while $Ca^{2+}$/CaM interaction switches channels between discrete high and low $P_O$ gating modes (*Ben-Johny et al., 2015*). How do allosteric regulators override CaM-signaling? First, stac and fhf use unique interfaces on the channel CI to selectively tune $Ca^{2+}$-feedback. Second, stac locks $Ca_V1$ into a high $P_O$ gating mode irrespective of whether apoCaM or $Ca^{2+}$/CaM is bound, effectively disengaging the pore from CaM-conformational changes. For $Na_V1$, despite fhf binding, CaM undergoes a profound $Ca^{2+}$-dependent rearrangement (*Wang et al., 2012*; *Wang et al., 2014*) suggesting that fhf does not prevent $Ca^{2+}$ binding to CaM or $Ca^{2+}$/CaM interaction with effector interfaces. Instead, like stac and $Ca_V1$, fhf may override CaM-dependent changes to $Na_V$, akin to a clutch disengaging power transmission in mechanical systems. As fhf elicits a change in apoCaM conformation (*Figure 7A*) (*Gabelli et al., 2014*; *Wang et al., 2012*), baseline gating of $Na_V$ may also be altered (*Goldfarb et al., 2007*; *Lou et al., 2005*). This parallelism between stac and fhf hints at a shared mechanism.

$Ca^{2+}$-binding proteins (CaBPs) (*Haeseleer et al., 2000*) also suppress CaM signaling to $Ca_V1$ (*Lee et al., 2002*; *Yang et al., 2006*). Mechanistically, CaBPs exploit a mixed allosteric scheme – at low concentrations, they engage distinct interfaces from CaM but at higher concentrations displace CaM (*Findeisen and Minor, 2010*; *Oz et al., 2013*; *Yang et al., 2014*). The existence of other regulatory proteins that curtail $Ca^{2+}$-feedback points to a general class of auxiliary regulators of CaM-signaling to targets beyond $Na_V1$ and $Ca_V1$. Identifying such molecular players is critical to understand how CaM signaling is orchestrated.

## Biological implications of stac modulation of $Ca_V1$

Stac1/2 isoforms are widely expressed in multiple brain regions, including both the hippocampus and the midbrain (*Nelson et al., 2013*; *Suzuki et al., 1996*). Our experiments hint at low basal stac2 expression in guinea pig ventricular cardiac myocytes, although previous studies have failed to detect stac2 in murine heart (*Nelson et al., 2013*). Further quantitative analysis will help establish ambient stac levels including species-specific differences and potential modulatory effects on cardiac function. Interestingly, endogenous $Ca_V1$ in both hippocampal and midbrain neurons (*Bazzazi et al., 2013*; *Oliveria et al., 2012*) as well as ventricular cardiac myocytes exhibit CDI. As all stac variants shunted CDI of $Ca_V1$ in HEK293, it is possible that stac function may be tightly regulated in native settings. One possibility is that stac abundance may be tuned developmentally (*Suzuki et al., 1996*), pathologically, or via interacting proteins (*Satoh et al., 2006*). For instance, the transcription factor, NFAT binds to an upstream promoter region of stac2 gene to upregulate stac2 expression in osteoclasts as well as during hypoxic conditions in neural stem cells (*Jeong et al., 2018*; *Moreno et al., 2015*). Physiologically, as $Ca_V1$ CDI is a potent homeostatic mechanism that prevents pathological $Ca^{2+}$-overload (*Dunlap, 2007*), a low concentration regime of stac may be advantageous. By modulating a subpopulation of $Ca_V1$, stac may circumvent homeostatic requirements to amplify local $Ca^{2+}$-signals via sustained $Ca^{2+}$ influx. The C1 and SH3 domains may serve as scaffolds to localize stac to specific signaling complexes (*Campiglio and Flucher, 2017*; *Cohen et al., 1995*; *Colon-Gonzalez and Kazanietz, 2006*). It is also possible that phosphorylation of stac may dynamically tune its function (*Huttlin et al., 2010*). Resolving these complexities may unveil mechanisms that tune $Ca_V$ function spatially and temporally.

In cardiac myocytes, CDI of $Ca_V1$ is a key factor for action potential duration (*Limpitikul et al., 2014*; *Mahajan et al., 2008*). Experimentally, this importance is inferred from prolongation of action potentials upon expression of mutant $CaM_{1234}$ (*Alseikhan et al., 2002*). Yet, constitutive CaM expression may yield nonspecific effects (*Hall et al., 2013*; *Wang et al., 2007*) that obscure the net contribution of $Ca_V1$ CDI (*Zhang et al., 2015*). Acute elevation of the U-domain bypasses these ambiguities and confirms a key role for $Ca_V1$ CDI for cardiac action potentials. Pathophysiologically, differential expression of stac2 has been reported in right ventricular heart failure, hinting at a potential role in calcium remodeling during heart failure (*di Salvo et al., 2015*).

Post-transcriptional modification of $Ca_V1.3$ generates an assortment of variants with modified carboxy-termini (*Bock et al., 2011*; *Huang et al., 2012*). The apoCaM affinities of these variants are such that CaM fluctuations may redistribute channels between populations lacking or endowed with

apoCaM (*Bazzazi et al., 2013*), evoking concomitant changes in maximal $P_O$ and CDI of $Ca_V1.3$ (*Adams et al., 2014*). Stac uniformly locks these variants into a high $P_O$ configuration incapable of CDI, thereby supporting reliable and persistent $Ca^{2+}$-influx in spite of CaM. Notably, functional effects of $Ca_V1.3$ alternative splicing have been shown to be cell-type specific suggesting that auxiliary regulators may tune channel properties (*Scharinger et al., 2015*). Fitting with these regulatory possibilities, disruption of stac modulation of $Ca_V1.3$ in Drosophila alters circadian rhythm (*Hsu et al., 2018*).

## Biological implications of fhf modulation of $Na_V1$

Unlike canonical fibroblast growth factors, fhf lack a secretory signal sequence (*Smallwood et al., 1996*) and serve as intracellular proteins (*Schoorlemmer and Goldfarb, 2001*). Four distinct fhf isoforms have been identified with tissue-specific expression in neurons, cardiomyocytes, and skeletal muscle (*Goldfarb, 2005*; *Kraner et al., 2012*; *Smallwood et al., 1996*). Functionally, fhf isoforms promote $Na_V1$ trafficking and fast inactivation (*Pablo and Pitt, 2016*). More specifically, fhf adjust steady-state voltage-dependence of inactivation (*Lou et al., 2005*), elicit a kinetically distinct long-term inactivation (*Dover et al., 2010*), and modify resurgent current (*Yan et al., 2014*). Our present findings suggest that fhf1 also tunes CDI of $Na_V1$. Physiologically, $Na_V$ CDI may be prominent during repetitive activity, as excess $Ca^{2+}$ accumulation may inhibit Na currents. Thus, suppression of $Na_V1$ CDI by fhf may enhance repetitive firing. Interestingly, loss of fhf1 and/or fhf4 result in diminished firing properties of cerebellar Purkinje neurons (*Bosch et al., 2015*; *Goldfarb et al., 2007*), while loss of fhf2 reduces cardiac conduction (*Park et al., 2016*; *Wang et al., 2011a*). It is possible that loss of fhf may enhance net CDI thus contributing to diminished excitability in these cells. As mutations in fhf1 are associated with epileptic encephalopathy (*For CENetDDD Study group‡* et al., 2016*) and cardiac conduction disorders (*Hennessey et al., 2013*) while mutations in fhf4 are linked to spinocerebellar ataxia (*Brusse et al., 2006*), resolving the dynamic interplay between CaM and fhf in tuning $Na_V1$ may be critical for understanding pathogenic mechanisms.

## New strategy for synthetic ion channel modulation

Finally, our results highlight the possibility of engineering synthetic regulation to tune CaM signaling. While $Ca_V1.3$ is insensitive to Mona SH3, insertion of an RxxK motif (*Harkiolaki et al., 2003*) into the carboxy-tail preIQ segment allows latent modulation by Mona SH3. Given the structural similarity of the CI modules of $Ca_V1$, $Ca_V2$, and $Na_V1$, and sequence variability within the preIQ domain, emerging protein engineering methods may be used to screen for synthetic modulators of related ion channel families. As the ligand specificity of SH3 domains can be custom-engineered (*Nguyen et al., 2000*) and subcellular localization tuned via targeting motifs (*Komatsu et al., 2010*), a custom library of synthetic regulators may be developed to combinatorially modify kinetic properties of $Ca_V1$, $Ca_V2$, or $Na_V1$ channels with spatiotemporal specificity. Generalizing this approach may lead to the development of new tools to manipulate $Ca^{2+}$ signaling.

In all, our findings unravel the elegant interplay between a novel class of allosteric regulators and CaM in orchestrating the activity of $Ca_V$ and $Na_V$ channels.

# Materials and methods

Key resources table

| Reagent type (species) or resource | Designation | Source or reference | Identifiers | Additional information |
|---|---|---|---|---|
| Gene (rat) | $\beta_{2A}$ | PMID: 1370480 | GenBank: M80545 | |
| Gene (rat) | $\alpha_2\delta$ | PMID: 8107966 | NCBI: NM_012919 | |
| Gene (*Oryctolagus cuniculus*) | $Ca_V1.2$ | PMID: 1718988 | NCBI: NM_001136522 | |
| Gene (rat) | $Ca_V1.3_S$ | PMID: 20139964 | GenBank: AF370009.1 | |
| Gene (rat) | $Ca_V1.3_{MQDY}$ | PMID: 24120865, 22284185 | | |

*Continued on next page*

*Continued*

| Reagent type (species) or resource | Designation | Source or reference | Identifiers | Additional information |
|---|---|---|---|---|
| Gene (human) | $Ca_V1.4_{43*}$ | PMID: 22069316 | | Laboratory of Dr. Soong Tuck Wah (National University of Singapore) |
| Gene (human) | $Ca_V2.1$ splice variant 37a(EFa) with $43^+/44^-/47^-$ | PMID: 12451115 | | |
| Gene (human) | $Ca_V2.2$ | PMID: 1321501, 10233069 | GenBank: M94172.1 | |
| Gene (rat) | $Ca_V2.3$ | PMID: 8388125, 18400181 | NCBI: NM_019294.2 | |
| Gene (rat) | $Na_V1.4$ | PMID: 2175278 | | |
| Gene (human) | stac1 | Origene | NCBI: NP_003140.1 | |
| Gene (mouse) | stac2 | Origene | NCBI: NP_666140.1 | |
| Gene (human) | stac3 | Origene | NCBI: NP_659501.1 | |
| Gene (human) | fhf | PMID: 8790420 | | Laboratory of Dr. Jeremy Nathans (Johns Hopkins University). |
| Gene (human) | Mona SH3 | PMID: 12773374 | | Synthesized by Genscript based on sequence in publication |
| Peptide (mouse) | U-peptide | this paper | | Peptide sequence KVDPVYETLRYGTSLALMNRSS synthesized by Genscript |
| Competent cells (*E. coli*) | DH5α | Invitrogen: 18265017 | | |
| Cell line (human) | HEK293 | other | RRID: CVCL_0045 | |
| Biological sample (guinea pig) | aGPVM | PMID: 24076394 | | Generated from Hartley strain guinea pigs |
| Antibody | anti-stac1 | Abcam: ab181157 | | 1:100 |
| Antibody | anti-stac2 | Abcam: ab156080 | | IHC – 1:100 WB – 1:250 |
| Antibody | anti-α-actinin | Sigma Aldrich: A7811 | RRID: AB_476766 | 1:300 |
| Antibody | goat anti- rabbit IgG Alexa Fluor 594 | Abcam: ab150080 | RRID: AB_2650602 | 1:1000 |
| Antibody | goat anti-mouse IgG1 Alexa Fluor 488 | Thermo Fischer: A21121 | RRID: AB_2535764 | 1:1000 |
| Antibody | Goat Anti-Rabbit IgG (H + L) | Jackson ImmunoResearch: 111-035-144 | RRID: AB_2307391 | 1:10,000 |
| Recombinant DNA reagent | $Ca_V1.3_L$ | PMID: 20139964 | | Engineered from $Ca_V1.3_S$ and human long distal carboxyl tail (NCBI: NM_000718) |
| Recombinant DNA reagent | $Ca_V2.3/1.3$ CI | PMID: 24441587 | | |
| Recombinant DNA reagent | $Ca_V1.3$- $CaM_{WT}$ | PMID: 24441587 | | |
| Recombinant DNA reagent | $Ca_V1.2$- $CaM_{WT}$ | PMID: 15087548 | | |

*Continued*

| Reagent type (species) or resource | Designation | Source or reference | Identifiers | Additional information |
|---|---|---|---|---|
| Recombinant DNA reagent | Na$_V$1.4-CaM | this paper | | Engineered by fusing CaM$_{WT}$ carboxy-tail of Na$_V$1.4 |
| Recombinant DNA reagent | Ca$_V$1.3$_{RxxK}$ | this paper | | Engineered from Ca$_V$1.3$_S$ |
| Recombinant DNA reagent | CFP-stac3 | this paper | | stac3 was cloned into CFP vector with *NotI* and *XbaI* |
| Recombinant DNA reagent | YFP-Ca$_V$1.3 CI | PMID: 23591884 | | |
| Recombinant DNA reagent | YFP-Ca$_V$1.3 PCI | PMID: 23591884 | | |
| Recombinant DNA reagent | YFP-Ca$_V$1.3 IQ | PMID: 23591884 | | |
| Recombinant DNA reagent | Ven-C1 | this paper | | stac2 C1 was cloned into Venus vector (PMID: 26997269) with *NotI* and *XbaI* |
| Recombinant DNA reagent | Ven-linker region | this paper | | stac2 linker region was cloned into Venus vector (PMID: 26997269) with *NotI* and *XbaI* |
| Recombinant DNA reagent | Ven-SH3-SH3 | this paper | | stac2 SH3-SH3 was cloned into Venus vector (PMID: 26997269) with *NotI* and *XbaI* |
| Recombinant DNA reagent | Ven-U-motif | this paper | | stac2 U-motif was cloned into Venus vector (PMID: 26997269) with *NotI* and *XbaI* |
| Recombinant DNA reagent | stac2 (KVD/AAA) | this paper | | Quickchange PCR with stac2 |
| Recombinant DNA reagent | stac2 (PVY/AAA) | this paper | | Quickchange PCR with stac2 |
| Recombinant DNA reagent | stac2 (ETL/AAA) | this paper | | Quickchange PCR with stac2 |
| Recombinant DNA reagent | stac2 (RYG/AAA) | this paper | | Quickchange PCR with stac2 |
| Recombinant DNA reagent | stac2 (TSL/AAA) | this paper | | Quickchange PCR with stac2 |
| Recombinant DNA reagent | stac2 (NRS/AAA) | this paper | | Quickchange PCR with stac2 |
| Recombinant DNA reagent | stac2 (S/A) | this paper | | Quickchange PCR with stac2 |
| Sequence-based reagent | Ven-C1 forward primer | this paper | | cttctcgcggccgc tatgaccgaa atga gcgagaa |
| Sequence-based reagent | Ven-C1 reverse primer | this paper | | tcagaattctagattat tgctggt gggagatctc |
| Sequence-based reagent | Ven-linker region forward primer | this paper | | cttctcgcggccgcta catctttt cgacgcaact |
| Sequence-based reagent | Ven-linker region reverse primer | this paper | | tcagaattctagatta gtacatg ggccccacg |
| Sequence-based reagent | Ven-SH3-SH3 forward primer | this paper | | cttctcgcggccgc ttcctacgt cgccctct |

*Continued on next page*

*Continued*

| Reagent type (species) or resource | Designation | Source or reference | Identifiers | Additional information |
|---|---|---|---|---|
| Sequence-based reagent | Ven-SH3-SH3 reverse primer | this paper | | tcagaattctagattat cagatctct gccaaggag |
| Sequence-based reagent | Ven-U-motif forward primer | this paper | | cttctcgcggccgctaagg tggac ccagtttatga |
| Sequence-based reagent | Ven-U-motif reverse primer | this paper | | tcagaattctagattag ctggaa cggttcatcag |
| Sequence-based reagent | stac2 (KVD/AAA) sense | this paper | | ctactgggaccagcgg ggcggcgg ccccagt ttatgagacgc |
| Sequence-based reagent | stac2 (KVD/AAA) antisense | this paper | | gcgtctcataaactggg gccgccgc cccgctgg tcccagtag |
| Sequence-based reagent | stac2 (PVY/AAA) sense | this paper | | ccagcgggaaggtggac gcagc tgctgagacgct gcgctatg |
| Sequence-based reagent | stac2 (PVY/AAA) antisense | this paper | | catagcgcagcgtctc agcagct gcgtccacc ttcccgctgg |
| Sequence-based reagent | stac2 (ETL/AAA) sense | this paper | | ggtggacccagttt atgcggcgg cgcgct atggcacctcc |
| Sequence-based reagent | stac2 (ETL/AAA) antisense | this paper | | ggaggtgccatagcgc gccgcc gcataaact gggtccacc |
| Sequence-based reagent | stac2 (RYG/AAA) sense | this paper | | cccagtttatgagacgc tggccgc tgccacctcc ctggcactgatg |
| Sequence-based reagent | stac2 (RYG/AAA) antisense | this paper | | catcagtgccaggg aggtggca gcggccagc gtctcataaactggg |
| Sequence-based reagent | stac2 (TSL/AAA) sense | this paper | | acgctgcgctatgg cgccgccgc ggcactga tgaaccg |
| Sequence-based reagent | stac2 (TSL/AAA) antisense | this paper | | cggttcatcagtgc cgcggcggc gccatag cgcagcgt |
| Sequence-based reagent | stac2 (NRS/AAA) sense | this paper | | gatgtgctgctga agctggcagcg gccatc agtgccagggaggtg |
| Sequence-based reagent | stac2 (NRS/AAA) antisense | this paper | | cacctccctggc actgatggccgc tgcc agcttcagcagcacatc |
| Sequence-based reagent | stac2 (S/A) sense | this paper | | cactgatgaacc gttccgccttc agcag cacatctg |
| Sequence-based reagent | stac2 (S/A) antisense | this paper | | cagatgtgctgctga aggcgg aacggttca tcagtg |
| Software, algorithm | PyMOL | http://www.pymol.org/ | RRID: SCR_000305 | |

## Molecular biology and peptide synthesis

$Ca_V1.2$, $Ca_V1.3$, $Ca_V1.4_{43*}$, $Ca_V2.1$, $Ca_V2.2$, $Ca_V2.3$, and $Na_V1.4$ variants were unmodified from previously published constructs: $Ca_V1.2$ (NM001136522) (*Wei et al., 1991*), $Ca_V1.2$-$CaM_{WT}$ (*Mori et al., 2004*), $Ca_V1.3_S$ (AF370009.1), $Ca_V1.3_L$ engineered from $Ca_V1.3_S$ and human long distal carboxyl tail (NM000718) (*Liu et al., 2010*), RNA-edited variant $Ca_V1.3_{MQDY}$ (*Bazzazi et al., 2013*; *Huang et al., 2012*), $Ca_V1.4_{43*}$ was gifted from Dr. Soong Tuck Wah (National University of Singapore), $Ca_V2.1$

splice variant 37a(EFa) with $43^+/44^-/47^-$ (*Soong et al., 2002*) was gifted from Dr. Terry Snutch (University of British Columbia), $Ca_V2.2$ (*Jones et al., 1999*), $Ca_V2.3$ (*Mori et al., 2008*), $Na_V1.4$ (*Trimmer et al., 1990*). Stac variants were purchased from Origene: human stac1 mRNA transcript 1 (NP003140.1), mouse stac2 (NP666140.1), and human stac3 isoform 2 (NP659501.1). U-peptide was synthesized by Genscript (KVDPVYETLRYGTSLALMNRSS). Fhf variants were gifted from Dr. Gordon Tomaselli and Dr. Jeremy Nathans (Johns Hopkins University).

## Cell culture and transfection of HEK293 cells

For whole-cell electrophysiology, single-channel electrophysiology, and immunohistochemistry, HEK293 cells (ATCC; mycoplasma tested negative) were cultured on glass coverslips in 10 cm dishes and transfected by a calcium phosphate method (*Peterson et al., 1999*) with the following amounts of DNA: 3 µg of SV40 T antigen to enhance expression, 2–8 µg of $\alpha_1$-subunit of $Ca^{2+}$ or $Na^+$ channel depending on expression, 8 µg from rat $\beta_{2A}$ (*Perez-Reyes et al., 1992*) (M80545), 8 µg from rat $\alpha_2\delta$ (*Tomlinson et al., 1993*) (NM012919.2), and 8 µg of the stac1, stac2, or stac3 variants indicated.

For FRET two-hybrid experiments, cells were cultured on glass-bottom dishes and transfected with a standard polyethylenimine protocol (*Lambert et al., 1996*). Epifluorescence measurements were recorded 1–2 days after transfection.

## Adult guinea pig ventricular myocyte isolation

Adult guinea pig ventricular myocytes (aGPVMs) were isolated from whole hearts of Hartley strain guinea pigs 3–4 weeks old (250–350 g). Guinea pigs were anesthetized via intraperitoneal injection with pentobarbital (35 mg/kg). Hearts were then excised, and single ventricular myocytes were isolated following a previously published protocol (*Joshi-Mukherjee et al., 2013*). Cells were plated on glass coverslips that were laminin (20 µg/mL) coated overnight at 4℃.

Immunohistochemistry aGPVMs plated on glass coverslips were first washed three times with cold PBS and then fixed in 3.7% paraformaldehyde (15710, Electron Microscopy Sciences) in PBS for 15 min. After washing three times with PBS, cells were permeabilized in cold 0.5% Triton X-100 in tris buffered saline (TBS) for 20 min and then blocked with 10% goat serum in PBS for 1 hr at room temperature. Cells were incubated overnight at 4℃ in primary antibodies diluted in antibody diluent solution (IW-1000, IHC World): monoclonal anti-$\alpha$-actinin (sarcomeric) antibody produced in mouse (1:300, A7811), anti-STAC (stac1) antibody [EPR12805]-N-terminal (1:100, ab181157) or anti-STAC2 (stac2) antibody-N-terminal (1:100, ab156080) produced in rabbit. Next day, cells were rinsed three times with 0.05% TWEEN20 (Sigma P9416) in TBS (TBS-T) for 5 min each. In the dark, cells were incubated with secondary antibodies (1:1000): goat anti-mouse IgG1 Alexa Fluor 488 (1:1000, A21121), goat anti-rabbit IgG Alexa Fluor 594 (1:1000), and DAPI (1:10000) diluted in antibody solution for 45 min at room temperature and then washed three times with TBS-T for 5 min each. Stained cells were mounted with prolong gold mounting media (Invitrogen) on a microscope slide (Fischer Scientific).

Transfected HEK293 were immunostained following a similar protocol to that of aGPVM, but were not labelled with sarcomeric primary antibody and its respective secondary antibody.

Western blot aGPVMs and HEK293 cells were washed twice with PBS buffer. Cells were harvested with 1 mL 1x RIPA buffer (20–188, Sigma Aldrich) containing half a tablet of complete mini-EDTA-free protease inhibitor (11836170001, Sigma Aldrich) and incubated at 4℃ for 30 min. Samples were centrifuged at 15,000 RPM for 15 min, and the pellet was discarded. Then, 2–5 µg of proteins in the supernatant were heated at 37℃ for 30 min with 2x Laemmli sample buffer (S3401, Sigma Aldrich). Samples were loaded into 4–12% gradient gel (NP0335BOX, Invitrogen) with PageRuler plus pre-stained protein ladder (26619, Invitrogen) and run at 100 V for 2 hr at room temperature in running buffer: 1x NuPAGE MOPS SDS running buffer: 50 mM MOPS, 50 mM Tris base, 0.1% SDS, 1 mM EDTA, pH to 7.7. Proteins were transferred on ice from the gel to nitrocellulose membrane (10600003, GE Healthcare Life science) for 75 min at 10 V in transfer buffer: 24 mM Tris base, 192 mM glycine, 20% v/v methanol. Membrane was blocked with 5% (w/v) Blotting-Grade-Blocker (1706404, Bio-Rad) in 1x TRIS-buffered saline for 1 hr at 4℃. Primary antibody for stac2 (1:250) was added to the blocking buffer with 0.1% (v/v) Tween 20 (1706404, Bio-Rad) and incubated overnight at 4℃. Next day, the membrane was washed three times for 5 min each with TBS with 0.1% (v/v) Tween 20 (TBS-T). The secondary antibody (111-035-144, Jackson ImmunoResearch; 1:10,000) was

added to the blocking buffer with 0.1% (v/v) Tween 20 and incubated for 1 hr. The membrane was washed again three times for 5 min each with TBS-T. Finally, western blots were developed with SuperSignal West Pico Chemiluminescent Substrate (34580, ThermoFischer) and images were collected on an Alpha InnoTech FluorChem HD2 imaging system.

## Confocal optical imaging

Images of immunostained tissue slices and cells were captured with either an Olympus Fluorview FV300 confocal laser scanning microscope or an LSM780 (Carl Zeiss, Oberkochen, Germany) confocal microscope. For the FV300, we used Fluoview software (Olympus) with a PlanApo 403 or 603 oil objective (NA 1.40, PLAPO60XO3; Olympus). Argon laser (488 nm) was used to excite Alexa Fluor 488 (green), and Helium Neon (HeNe) Green Laser was used to excite Alexa Fluor 594 (red). Olympus optical filters used were 442/515 nm excitation splitter (FV-FCV), 570 nm emission splitter (FV-570CH), BA510 IF and BA530RIF for green emission channel, and 605 BP filter for red channel. Images were processed in ImageJ. Similar settings were used for the LSM780 setup.

## Whole-cell electrophysiology

Whole-cell voltage-clamp recordings for HEK293 were collected at room temperature 1–2 days after transfection with Axopatch 200A (Axon Instruments). Glass pipettes (BF150-86-10, Sutter Instruments) were pulled with a horizontal puller (P-97; Sutter Instruments Company) and fire polished (Microforge, Narishige, Tokyo, Japan) to have 1–3 MΩ resistance. Recordings were low-pass filtered at 2 kHz and sampled at 10 kHz with P/8 leak subtraction and 70% series resistance and capacitance compensation. For recordings of $Ca_V1.2$ (*Figure 1A–B*, *Figure 5I–K*, *Figure 6B*), $Ca_V1.3_S$ (*Figure 1C–D*, *Figure 3—figure supplement 1A–B and and G*, *Figure 7D–E*, and *Figure 8B–C*), $Ca_V1.4_{43*}$ (*Figure 1E–F*), $Ca_V2.2$ (*Figure 1I–J*), $Ca_V2.3/1.3$ CI chimera (*Figure 2E–F*), $Ca_V1.3$-CaM (*Figure 3A–B*, *Figure 3—figure supplement 1C–F*, and *Figure 5G–H*), $Ca_V1.2$-CaM (*Figure 2C–D*, *Figure 5A–F*) and $Ca_V1.3_{RxxK}$ (*Figure 8D–E*) exogenously expressed in HEK293 cells, the internal solution contained (in mM): $CsMeSO_3$, 114; $CsCl_2$, 5; $MgCl_2$, 1; MgATP, 4; HEPES, 10; BAPTA, 10; adjusted to 295 mOsm with $CsMeSO_3$ and pH 7.4 with CsOH. The external solution contained (in mM): $TEA-MeSO_3$, 140; HEPES, 10; $CaCl_2$, or $BaCl_2$ 40; adjusted to 300 mOsm with $TEA-MeSO_3$ and pH 7.4 with TEA-OH. For recordings of $Ca_V2.1$ (*Figure 1G–H*) and $Ca_V2.3$ (*Figure 1K–L*), the internal solution contained (in mM): $CsMeSO_3$, 135; $CsCl_2$, 5; $MgCl_2$, 1; MgATP, 4; HEPES, 10; EGTA, 1; adjusted to 295 mOsm with $CsMeSO_3$ and pH 7.4 with CsOH. The external solution contained (in mM): $TEA-MeSO_3$, 140; HEPES, 10; $CaCl_2$, or $BaCl_2$ 5; adjusted to 300 mOsm with $TEA-MeSO_3$ and pH 7.4 with TEA-OH. At a holding potential of $-80$ mV, we used a family of test pulses from $-30$ mV to +50 mV with repetition intervals of 20 s. Custom MATLAB (Mathworks) software (https://github.com/manubenjohny/WCDTY; copy archived at https://github.com/elifesciences-publications/WCDTY) was used to determine peak current and fraction of peak current remaining after either 300 ms ($r_{300}$) or 800 ms ($r_{800}$) of depolarization.

We incubated aGPVMs for 20–48 hr after isolation in 5 µM ryanodine for 5–10 min before we collected whole-cell recordings. The internal recording solution contained (in mM) $CsMeSO_3$, 114; $CsCl_2$, 5; $MgCl_2$, 1; MgATP, 4; HEPES, 10; BAPTA, 10; ryanodine, 0.005 adjusted to 295 mOsm with $CsMeSO_3$ and pH 7.4 with CsOH. Cells were sealed in Tyrodes solution, which contained (in mM): NaCl, 135; KCl, 5.4; $CaCl_2$, 1.8; $MgCl_2$, 0.33; $NaH_2PO_4$, 0.33; HEPES, 5; glucose, 5 (pH 7.4). For CDI measurements, external solutions containing (in mM): $TEA-MeSO_3$, 140; HEPES, 10; $CaCl_2$, or $BaCl_2$ 40; adjusted to 300 mOsm with $TEA-MeSO_3$ and pH 7.4 with TEA-OH were perfused. Welch's T-test was used to verify statistical significance among the population data.

For CDI recordings, we determined required sample size based on power analysis. Based on historical estimates of normal variation in CDI/CDF measurements, we computed the sample size required such that type I and type II errors are 5% to be 3.5. Thus, we obtained at least four independent measurements for all electrophysiological experiments.

Current-clamp recordings of aGPVMs were performed on the same setup and were filtered at 5 kHz and sampled at 25 kHz. The internal solution contained (in mM): K glutamate, 130; KCl, 9; NaCl, 10; $MgCl_2$, 0.5; EGTA, 0.5; MgATP, 4; HEPES, 10; adjusted to pH 7.3 with KOH. The external solution contained (in mM): NaCl, 135; KCl, 5.4; $CaCl_2$, 1.8; $MgCl_2$, 0.33; $NaH_2PO_4$, 0.33; HEPES, 5; glucose, 5 (pH 7.4). The time from upstroke to 80% repolarization ($APD_{80}$) was measured with MATLAB

(Mathworks) and used as a metric for comparing physiological output between peptide treated and untreated. For experiments with U-peptide, peptide was dissolved in ddH$_2$O to 2 mg/mL and then diluted to 500 µM in the appropriate internal solution.

## Single-channel electrophysiology

Single-channel recordings were performed at room temperature using an on-cell configuration previously established in the laboratory (*Tay et al., 2012*) with the same setup as used for whole-cell electrophysiology. Glass pipettes were pulled and polished from ultra-thick-walled borosilicate glass (BF200-116-10, Sutter Instruments) and coated with sylgard to have 5–10 MΩ resistance. Recordings were filtered at 2–5 kHz. The pipette solution contained (in mM): TEA-MeSO$_3$, 140; HEPES, 10; BaCl$_2$ 40; adjusted to 300 mOsm with TEA-MeSO$_3$ and pH 7.4 with TEA-OH. The external solution contained (in mM): K glutamate, 132; KCl, 5; NaCl, 5; MgCl$_2$, 3; EGTA, 2; HEPES, 10; adjusted to 300 mOsm with glucose and pH 7.4 with KOH. Cell-attached single-channel currents were measured during 200 ms voltage ramps between −80 and +70 mV (portions between −50 and 40 mV displayed and analyzed) as previously described. For each patch, we recorded 80–150 sweeps with a repetition interval of 12 s. Patches were analyzed as follows: (1) The leak for each sweep was fit and subtracted from each trace. (2) The unitary current relation, $i(V)$, was fit to the open-channel current level using the following equation:

$$i(V) = -g \cdot (V - V_S) \cdot exp(-(V - V_S) \cdot z \cdot F/(R \cdot T)) \,/\, (1 - exp(-(V - V_S) \cdot z \cdot F/(R \cdot T)))$$ where $g$ is the single-channel conductance (~0.2 pA/mV), $z$ is the apparent valence of permeation (~2.1), $F$ is Faraday's constant, $R$ is the gas constant, and $T$ is the temperature in degrees Kelvin (assumed room temperature). These parameters were held constant for all patches, except for slight variations in the voltage-shift parameter $V_s$ ~ 35 mV, as detailed below. (3) All leak-subtracted traces for each patch were averaged (and divided by the number of channels in the patch) to yield an I–V relation for that patch. As slight variability in $V_S$ was observed among patches, we calculated an average $V_S$ for each construct, $V_{S,AVE}$. The data from each patch were then shifted slightly in voltage by an amount $\Delta V = V_{S,AVE} - V_S$, with $\Delta V$ typically about ±5 mV. This maneuver allowed all patches for a given construct to share a common open-channel GHK relation. Thus shifted, the I–V relations obtained from different patches for each condition/construct were then averaged together. (4) $P_O$ at each voltage was determined by dividing the average $I$ (determined in step three above) into the open-channel GHK relation. Channel number was determined by the maximal number of overlapping opening events upon application of the channel agonist Bay K8644 (5 µM) at the end of each recording. For modal analysis, a dashed line discriminator was chosen to be the average single-trial $P_O$ = 0.075 such that traces with average single-trial $P_O$ >0.075 were categorized as high $P_O$ while the remaining traces were considered to be low $P_O$.

## Quantitative calcium photo-uncaging

All Ca$^{2+}$-uncaging experiments were conducted on a Nikon TE2000 inverted microscope with a Plan Fluor Apo 40 × oil objective as previously described (*Ben-Johny et al., 2014*). Briefly, a classic Cairn UV flash photolysis system was used for Ca$^{2+}$-uncaging with brief UV pulses of ~1.5 ms in duration powered by a capacitor bank of up to 4000 µF charged to 200–290V. For concurrent Ca$^{2+}$ imaging, Fluo4FF and Alexa568 dyes were dialyzed via patch pipette and imaged using Argon laser excitation (514 nm). Background fluorescence for each cell was measured prior to pipette dialysis of dyes and subtracted subsequently. A field-stop aperture was used to isolate fluorescence from individual cells. Dual-color fluorescence emission was attained using a 545DCLP dichroic mirror, paired with a 545/40 BP filter for detecting Fluo4FF, and a 580LP filter for detecting Alexa568. Typically, uncaging experiments were conducted after ~2 min of dialysis of internal solution. Welch's T-test was used to verify statistical significance between the population data.

For all Ca$^{2+}$-uncaging experiments, the internal solution contained (in mM): CsMeSO$_3$, 120; CsCl, 5; HEPES (pH 7.4 with CsOH), 10; Fluo-4FF pentapotassium salt (Invitrogen), 0.01; Alexa 568 succinimidyl ester (Invitrogen), 0.0025; Citrate, 1; DM-Nitrophen EDTA (DMN) and CaCl$_2$ were adjusted to obtain the desired Ca$^{2+}$ flash. Typically, for flashes in the range 0.5–2 µM, DMN, 1 mM; and CaCl$_2$, 0.7 mM. For the 2–8 µM range, DMN, 2 mM; and CaCl$_2$, 1.4 mM. For larger Ca$^{2+}$ steps, DMN, 4 mM; and CaCl$_2$, 3.2 mM. As DMN can bind Mg$^{2+}$, all experiments were conducted with 0 mM Mg$^{2+}$

internally. For all Na channel experiments, the bath solution contained (in mM): TEA-MeSO$_3$, 45; HEPES (pH 7.4), 10; NaCl, 100; at 300 mOsm, adjusted with TEA-MeSO$_3$.

## FRET-two-hybrid assay

To collect a range of donor molecule ($D_{free}$) concentrations, HEK293 cells were transfected with combinations of DNA ratios. Cells were immersed in 2 mM Ca$^{2+}$ Tyrodes solution, which contained (in mM): NaCl, 138; KCl, 4; CaCl$_2$, 2; MgCl$_2$, 1; HEPES, 10; glucose, 10. Three-cube FRET fluorescence measurements were performed under resting Ca$^{2+}$ concentrations on an inverted fluorescence microscope. FRET efficiency ($E_A$ and $E_D$) was calculated for each cell (*Erickson et al., 2001*) and a binding curve, either $E_A = [D_{free}]/(K_{d,EFF} + [D_{free}]) \cdot E_{A,max}$ or $E_D = [A_{free}]/(K_{d,EFF} + [A_{free}])$, was fit to compute the effective dissociation constant ($K_{d,EFF}$).

## Acknowledgements

We thank Deborah DeSilvestre, Rebeca Joca, and Travis Babola for assistance with immunohistochemistry, Hikeki Nakamura for confocal microscope training, Trudeau lab (Ashley Johnson and Sara Coddin) and Colecraft lab (Travis Morgenstern and Scott Kanner) for western blotting expertise. We are grateful for insightful discussions with the Calcium Signals Laboratory and Inoue Synthetic Biology Lab. Finally, we are indebted to the inspiration of Dr. David T Yue, who taught us to pursue science with a passion for the truth. This work was supported by grants from NINDS (DTY, IED, TI), NIMH (DTY, MBJ), NHLBI (GFT) and NSF (JN).

## Additional information

### Funding

| Funder | Author |
| --- | --- |
| National Science Foundation | Jacqueline Niu |
| National Institute of Neurological Disorders and Stroke | Ivy E Dick<br>David T Yue<br>Takanari Inoue |
| National Institute of Mental Health | David T Yue<br>Manu Ben-Johny |
| National Heart, Lung, and Blood Institute | Gordon Tomaselli<br>Manu Ben-Johny |

The funders had no role in study design, data collection and interpretation, or the decision to submit the work for publication.

### Author contributions

Jacqueline Niu, Conceptualization, Data curation, Formal analysis, Investigation, Visualization, Methodology, Writing—original draft, Writing—review and editing; Ivy E Dick, Resources, Data curation, Funding acquisition, Writing—review and editing; Wanjun Yang, Resources; Moradeke A Bamgboye, Data curation; David T Yue, Funding acquisition; Gordon Tomaselli, Resources, Funding acquisition, Writing—review and editing; Takanari Inoue, Conceptualization, Supervision, Writing—review and editing; Manu Ben-Johny, Conceptualization, Data curation, Formal analysis, Investigation, Visualization, Writing—original draft, Writing—review and editing

### Author ORCIDs

Manu Ben-Johny http://orcid.org/0000-0002-5645-0815

### Ethics

Animal experimentation: This study was performed in strict accordance with the recommendations in the Guide for the Care and Use of Laboratory Animals of the National Institutes of Health. All of the animals were handled according to approved institutional animal care and use committee (IACUC) protocols of the Johns Hopkins University (GP15M172). The protocol was approved by the

Committee on the Ethics of Animal Experiments of the Johns Hopkins University. All surgery was performed under sodium pentobarbital anesthesia, and every effort was made to minimize suffering.

## Decision letter and Author response

Decision letter https://doi.org/10.7554/eLife.35222.024
Author response https://doi.org/10.7554/eLife.35222.025

## Additional files

### Supplementary files

• Transparent reporting form
DOI: https://doi.org/10.7554/eLife.35222.022

### Data availability

All data generated or analysed during this study are included in the manuscript and supporting files.

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
