## [Decision Letter]

Thank you for submitting your article "Allosteric regulators selectively shunt Ca^2+^-feedback of Ca_V_ and Na_V_ channels" for consideration by *eLife*. Your article has been reviewed by three peer reviewers, and the evaluation has been overseen by a Reviewing Editor and Richard Aldrich as the Senior Editor. The following individuals involved in review of your submission have agreed to reveal their identity: Bruce P Bean (Reviewer #2); Dejian Ren (Reviewer #3).

The reviewers have discussed the reviews with one another and the Reviewing Editor has drafted this decision to help you prepare a revised submission.

Summary:

This study describes two classes of proteins, stac and fhf, interfere with CaM binding to voltage-dependent Ca and Na channels, both of which have a CaM binding domain that stacs and fhfs can also bind, thereby reducing Ca-dependent inactivation. The data offer a potential explanation for why CaM can have targeted effects, and reveal interactions among channel modulators that have posed long-standing puzzles, e.g. why CaM sometimes has effects on channels with CaM binding domains but sometimes not.

Essential revisions:

1) The primary results on how stac modulates Ca channels are all with overexpression. Consequently, it is unclear to what extent normal expression levels participate in regulation of currents and what the nature of the competition (or not) with CaM is. Additional approaches to clarify these points are suggested by the reviewers.

2) The immunostaining in Figure 1 of stac expression in cardiac neurons is not convincing. The reviewers suggest either removal of these data or a more extensive validation of the antibodies and/or other means of verifying the signal.

3) A number of discrepancies with the published literature were noted. These should be acknowledged and discussed, and where possible, resolutions proposed.

4) The caliber of the data is high but the highly dramatic, hyperbolic language detracts from the overall quality of the manuscript. While some stylistic freedom is appropriate, it must not interfere with accuracy and readability. A few specific sentences are noted in the reviews, but the full manuscript should be revised substantially with this idea in mind.

The basis for these comments, in the reviewers' words, is included below, under "Major Comments" to facilitate the revision.

Title: It is not clear that the prevention of CaM modulation of channels by stac is meaningfully described by the word "shunt." Possibly a more straightforward work like "prevent" or "interferes with" would be more appropriate.

Major comments (in the reviewers' words; explaining basis for the essential revisions requested):

1) Interpretations from overexpression. A general issue concerning the significance is that the key results in how stac proteins modify behavior of calcium channels are all with overexpression of stac, and there is no evidence that there is any "tuning" from this mechanism physiologically (in stark contrast to previous data with stac3 in skeletal muscle, where the physiological relevance is quite clear). Previous studies in both heart and neurons show prominent calcium-dependent inactivation, giving little reason to think that stac inhibition of this effect is important. The author's experiments here disrupting the interaction also suggest little or no physiological significance in cardiac muscle. An RNAi or knock-out approach would be better suited for testing physiological relevance for native Ca_v_1.2 or Ca_v_1.3 channels.

Also, the authors conclude that STAC doesn't compete with CaM to suppress CDI, a mechanism apparently contrast to the "competition-based" mechanism implicated by Campiglio et al., 2018. While the multiple experiments in this manuscript do suggest that the authors' model is more convincing, a major evidence the authors used to "explicitly rule out" that a displacement of CaM binding by STAC is necessary for CDI suppression is by fusing CaM to Ca_V_'s C-terminus to increase local CaM concentration (Figure 3F-I). However, such a fusion still can't rule out that overexpressing STAC (as done in the paper) displaces CaM's binding to the site. One way to solve this problem is perhaps by directly monitor CaM binding with and without STAC. An easier, though less direct, experiment is perhaps to overexpress CaM (as done in the single channel recordings in Figure 4) and to test whether STAC still eliminates CDI when CaM binding is presumably saturated.

2) Immunostaining. A potentially novel result is the immuno evidence for stac2 in cardiac muscle. Previous work concluded stac3 is expressed in skeletal muscle and stac1 and stac2 in neurons, with no clear expression of any in cardiac muscle. But here the staining for stac2 is weak and there is no validation of the antibody, so this is very weak evidence -especially since the disruption experiment suggests no basal modulation that might suggest endogenous stac expression. Everyone who has worked with antibodies knows how easy it is to get weak off-target staining. The title of Figure 1 "Stac2 is differentially expressed in cardiac myocytes and neurons" is also somewhat misleading, as the experiments don't examine stac2's expression in neurons and the immunostaining data is not quantitative.

3) Relation to other work. The authors report several interesting findings in this paper. First, they found that all the STACs (STAC1-3) can abolish/reduce the calmodulin (CaM)-mediated CDI of L-type (Ca_V_1 sub-family) Ca_V_s (a finding also recently reported by other groups, e.g. Polster et al., 2015, Campiglio et al., 2018). Second, the authors used the FRET protein-interaction assay and found that STAC3 interacts with the proximal Ca^2+^-inactivating (PCI) segment of the Ca^2+^-inactivating (CI) module in the C-terminus of Ca_V_1.3, but not with the other segments, including the IQ motif immediately C-terminal to the PCI and the II/III loop previously found by others to interact with STACs in the Ca_V_1.1 (Yuen et al., 2017, Polster et al., 2018) or Ca_V_1.2 (Campiglio et al., 2018). Intriguingly, the authors found that a short "linker" between the C1 and the SH3 regions on STAC is required for STAC's suppression of CDI, and infusion of a U-motif peptide derived from this linker region is sufficient to remove CDI. This finding again is apparently contrast with a previous conclusion by others that the C1 domain is the major CDI-slowing determinant.

Mechanistically, the authors conclude that STACs allosterically control CDI and "lock" the channel into a high opening probability state, a mechanism contrast to a previously proposed one by which STAC competes with CaM. Interestingly, the authors also discovered that the regulation of Na_V_'s CDI by FGF has a structural requirement analogous to that of the regulation of Ca_V_s by STACs. Perhaps the most remarkable finding in the paper is that the authors were able to introduce an artificial Mona SH3-binding sequence in the PCI region to confer complete CDI inhibition by Mona SH3, a protein that doesn't affect the wild-type Ca_V_. Overall, there are many apparent "discrepancies" between the current paper and the two PNAS papers (Yuen et al., 2017, Campiglio et al., 2018), ranging from the STAC binding sites on Ca_V_s (PCI in this paper vs. the IQ motif and II/III loop in the others), and the important segment on STAC (linker vs. C1) to the mechanism (allosteric vs. competitive). The authors might want to elaborate the differences and provide potential explanations.

4) Writing. A comment on style: The authors use a prose style with so many dramatic statements and intensifying adjectives that it will likely irritate and even alienate some readers. In my opinion, a more sober presentation in a standard style that lets the results speak for themselves is more effective. Insisting on telling the reader how important the results are (like repeatedly using "Remarkably" for results that seem very straightforward, like seeing similar effects on Ca_v_1.4 as for Ca_v_1.1, Ca_v_1.2, and Ca_v_1.3) can make the reader think that he or she is receiving a sales pitch rather than being told results. There are too many examples of over-dramatic language to list but one was the statement that stac and fhf exert "orthogonal" control because one interacts with calcium channels and one with sodium channels. Calmodulin interacts with dozens if not hundreds of proteins, and of course there are ways that the interactions are regulated differently – is each one of these an orthogonal dimension? I had a similar reaction to sentences like "Synergistic analysis of the two proteins delineate mechanisms that confer personalized modulation and inform upon physiological consequences" and "Repurposing this modulatory principle furnishes a general strategy for engineering synthetic modulators that patently switch CaV and NaV channel feedback." (These sentences lie between a lack of clear meaning and inaccuracy.)

[Editors' note: further revisions were requested prior to acceptance, as described below.]

Thank you for resubmitting your work entitled "Allosteric regulators selectively prevent Ca^2+^-feedback of Ca_V_ and Na_V_ channels" for further consideration at *eLife*. Your revised article has been favorably evaluated by Richard Aldrich (Senior Editor), a Reviewing Editor, and two reviewers.

The manuscript has been improved but there are some important issues that need to be addressed, as outlined below:

One reviewer (whose comments are included verbatim below) identified some inconsistencies in the match between figure panels cited in the text and in the figures, and also noted that the control plots in the last two supplementary figures (Figure 7—figure supplement 1 and Figure 8—figure supplement 1) appear to be identical, but the duplication (if these are indeed duplicates) is neither justified nor indicated in the legends. The same panel appears to be duplicated in Figure 1—figure supplement 1 and Figure 3—figure supplement 1, although in the latter it is labeled as N=6 instead of N=9 as in the other three panels. These issues raise concern about the care with which the manuscript was assembled and presented.

Before a final decision is rendered, please ensure (1) that the correct data and figure panels are present throughout the manuscript, (2) that they are called correctly, (3) that any duplication of panels to facilitate comparison is clearly indicated as such (with appropriate justification for a common control group), and (4) that there are no other related errors. In your revision, please clearly indicate the corrections that have been made.

Reviewer comments:

The revision generally addresses the main issues raised on the original version. The data are extensive and comprise a very comprehensive picture of the molecular interactions with the multiple STAC proteins with multiple calcium channels. It is still unproven that the STAC interaction functions physiologically to "tune" Ca-dependent channel inactivation, but the authors make a good point that testing this via RNAi would be technically challenging given current experimental systems and making knock-out mice would be a major project. The fact that STAC proteins are heavily expressed in the brain gives some credence to the idea that they could be functionally important in neurons, or knock-outs.

The authors need to proof-read the manuscript more carefully. The reviewer copy was hard to read, as it consisted of a pdf incorporating cross-outs and additions, which made spotting errors difficult. However, it is obvious that not all changes in re-numbering figure references were done correctly (e.g. in the first paragraph of the subsection “Stac selectively suppresses Ca^2+^-feedback of Ca_V_1 channels”, Figure 2A should now be 1A, and in the second paragraph of the subsection “Stac interacts with CaV1 CI module to elicit CDI suppression”, Figure 3B should be 2B).

Also the fact that the figures were not numbered made the reviewing process more difficult than was necessary. Also, at the end of the manuscript there are supplementary figures that are exact duplicates.

---

## [Author Response]

Essential revisions:1) The primary results on how stac modulates Ca channels are all with overexpression. Consequently, it is unclear to what extent normal expression levels participate in regulation of currents and what the nature of the competition (or not) with CaM is. Additional approaches to clarify these points are suggested by the reviewers.2) The immunostaining in Figure 1 of stac expression in cardiac neurons is not convincing. The reviewers suggest either removal of these data or a more extensive validation of the antibodies and/or other means of verifying the signal.3) A number of discrepancies with the published literature were noted. These should be acknowledged and discussed, and where possible, resolutions proposed.4) The caliber of the data is high but the highly dramatic, hyperbolic language detracts from the overall quality of the manuscript. While some stylistic freedom is appropriate, it must not interfere with accuracy and readability. A few specific sentences are noted in the reviews, but the full manuscript should be revised substantially with this idea in mind.The basis for these comments, in the reviewers' words, is included below, under "Major Comments" to facilitate the revision. Title: It is not clear that the prevention of CaM modulation of channels by stac is meaningfully described by the word "shunt." Possibly a more straightforward work like "prevent" or "interferes with" would be more appropriate.

We thank the reviewers for this suggestion. The revised title replaces the word shunt with prevent.

Major comments (in the reviewers' words; explaining basis for the essential revisions requested):

1) Interpretations from overexpression. A general issue concerning the significance is that the key results in how stac proteins modify behavior of calcium channels are all with over-expression of stac, and there is no evidence that there is any "tuning" from this mechanism physiologically (in stark contrast to previous data with stac3 in skeletal muscle, where the physiological relevance is quite clear). Previous studies in both heart and neurons show prominent calcium-dependent inactivation, giving little reason to think that stac inhibition of this effect is important. The author's experiments here disrupting the interaction also suggest little or no physiological significance in cardiac muscle. An RNAi or knock-out approach would be better suited for testing physiological relevance for native Ca_v_1.2 or Ca_v_1.3 channels.Also, the authors conclude that STAC doesn't compete with CaM to suppress CDI, a mechanism apparently contrast to the "competition-based" mechanism implicated by Campiglio et al., 2018. While the multiple experiments in this manuscript do suggest that the authors' model is more convincing, a major evidence the authors used to "explicitly rule out" that a displacement of CaM binding by STAC is necessary for CDI suppression is by fusing CaM to Ca_V_'s C-terminus to increase local CaM concentration (Figure 3F-I). However, such a fusion still can't rule out that over-expressing STAC (as done in the paper) displaces CaM's binding to the site. One way to solve this problem is perhaps by directly monitor CaM binding with and without STAC. An easier, though less direct, experiment is perhaps to over-express CaM (as done in the single channel recordings in Figure 4) and to test whether STAC still eliminates CDI when CaM binding is presumably saturated.

We agree with the reviewers that mammalian knockout models of stac1 and stac2 would be powerful approaches to dissect the contribution of this modulatory mechanism to cardiac and neuronal function. However, to the best of our knowledge, there are no such models currently available and generation of one would be outside the scope of the present manuscript. For cardiac myocytes, two complexities further obscure analysis of stac modulation of Ca_V_ channels. First, the primary consequence of a reduction in Ca^2+^ channel inactivation is a prolongation of the phase 2 of the action potential. However, as mice exhibit minimal phase 2, murine models can be limiting in studying consequences of deficits in Ca_V_ inactivation. We chose freshly dissociated guinea pig models for pipette dialysis of U-motif peptides for this reason. Second, cultures of cardiac myocytes from guinea pig (and other organisms) de-differentiate within 24-48 hrs, resulting in marked changes in gene expression and a loss of channel modulation (e.g. β-adrenergic effect). This short time-window limits the practicality of approaches such as RNAi suppression. Given these limitations, we believe that the suggested experiments would be an exciting follow up study. Instead, we here focus on establishing a unified mechanistic framework for stac modulation of Ca_V_ channels and, in so doing, set the stage for subsequent in depth physiological analysis.

To ascertain stac levels necessary for physiological modulation, we undertook a live-cell FRET 2-hybrid assay to estimate holo-channel affinity for stac. We co-expressed YFP-tagged Ca_V_1.3 with CFP-tagged stac3 and measured FRET efficiencies from individual cells. Thus probed, we obtained an effective holochannel affinity of *K*_d,EFF_ = 1458 ± 251 *D*_free_ units ~ 47 nM. By comparison, similar holochannel analysis of CaM binding revealed *K*_d,EFF_ = 700 *D*_free_ units ~ 22 nM (Yang et al., 2014). Given this nanomolar affinity estimate, our expectation is that even low concentrations of stac would suffice to elicit functional modulation. Moreover, small changes in Ca_V_1 CDI elicits marked changes in cardiac action potential waveform suggesting that low levels of stac may be functionally important. These findings are shown in Figure 2.

In skeletal muscle where endogenous stac is very high (~ μM range), the vast majority of channels would already be stac-bound and therefore insensitive to any fluctuations. By contrast, if ambient stac levels were in the nanomolar range, then even small fluctuations would tune baseline *P*_O_ and CDI. Recent work shows that the transcription factor NFAT binds to an upstream promoter region of stac2 gene and upregulates stac2 expression in osteoclasts (Jeong et al., 2018) and during hypoxic conditions in neural stem cells (Moreno et al., 2015). As NFAT signaling is upregulated during pathological cardiac hypertrophy, such conditions may also enhance stac expression.

The reviewers further raise an important concern regarding the nature of interplay between stac and CaM in modulating Ca_V_1. Though experiments using tethered CaM demonstrates that stac is capable of inhibiting CDI despite the overwhelming local concentration, it is possible that stac may yet displace CaM from its interaction site. To address this concern, we used FRET 2-hybrid assay to monitor CaM binding to its primary interface for Ca_V_1.3, the CI domain, both in the presence and absence of stac. Figure 3 shows robust baseline interaction of CFP-tagged CaM to YFP-tagged Ca_V_1.3 CI with a relative dissociation constant, *K*_d,EFF_ ~ 4000 ± 291 *D*_free_ units. If stac competes with CaM for a common interface, then the relative affinity would be reduced yielding a larger value for *K*_d,EFF_. Indeed, co-expression of untagged CaM_1234_ with aforementioned FRET pairs resulted in ~11-fold reduction in apparent affinity, with *K*_d,EFF_ = 47153 ± 4815 *D*_free_ units. By contrast, co-expression of untagged-stac3 did not appreciably perturb CI-CaM interaction with *K*_d,EFF_ = 4182 ± 330. These results demonstrate that stac does not displace CaM from its binding interface. These results have been added to Figure 3.

In addition to the FRET experiments, we also co-expressed Ca_V_1.3_S_ with both CaM and stac2 as suggested by the reviewers. This maneuver also showed a reduction in CDI confirming that stac eliminates CDI even when CaM binding is saturated. The data is shown in Figure 3—figure supplement 2G.

2) Immunostaining. A potentially novel result is the immuno evidence for stac2 in cardiac muscle. Previous work concluded stac3 is expressed in skeletal muscle and stac1 and stac2 in neurons, with no clear expression of any in cardiac muscle. But here the staining for stac2 is weak and there is no validation of the antibody, so this is very weak evidence -especially since the disruption experiment suggests no basal modulation that might suggest endogenous stac expression. Everyone who has worked with antibodies knows how easy it is to get weak off-target staining. The title of Figure 1 "Stac2 is differentially expressed in cardiac myocytes and neurons" is also somewhat misleading, as the experiments don't examine stac2's expression in neurons and the immunostaining data is not quantitative.

We thank the reviewers for this suggestion.

To assess our ability to identify stac2 in immunostaining experiments, we undertook quantitative analysis of fluorescence intensities from confocal images of individual HEK293 cells expressing recombinant stac1-3 isoforms and probed via stac1 (top row) and stac2 (bottom row) antibodies. Black line in Figure 6—figure supplement 1B shows the distribution of fluorescence intensities from ~ 80 untransfected cells probed using anti-stac1 (top) with mean fluorescence intensities of F- = 218 ± 1.85 (s.e.m) a.u. HEK293 cells expressing recombinant stac1 when probed with anti-stac1 revealed a population of cells with enhanced fluorescence intensity with mean F- = 507 ± 77.63 (s.e.m) a.u. By comparison, fluorescence intensity distributions of HEK293 expressing recombinant stac2 or stac3 probed with anti-stac1 were indistinguishable from the distribution obtained for untransfected cells. In like manner, probing untransfected HEK293 cells with anti-stac2 antibody revealed minimal baseline staining with mean fluorescence intensity of F- = 266.8 ± 3.46. Exogenous expression of stac2 reveals a population of cells with enhanced fluorescence and mean fluorescence intensity, F-= 822.05 ± 143.9. By contrast, anti-stac2 labeling of cells expressing stac1 and stac3 revealed low fluorescence intensities comparable to the distribution for untransfected cells. These findings highlight the ability of the two antibodies to detect stac1 and stac2 respectively. Analysis of aGPVM revealed endogenous stac2 but not stac1.

As an alternative approach, we used western blot analysis to probe the presence of endogenous stac2 in cardiac myocytes. Analysis of untransfected HEK293 cells showed no signal when probed with stac2 antibody. However, analysis of cell lysates from HEK293 transfected with stac2 shows ~ 50 kDa band corresponding to stac2 (m.w. 47 kDa). Similarly, analysis of lysates from freshly dissociated cardiac myocytes show a similar-sized band as with stac2-transfected HEK cells.

3) Relation to other work. The authors report several interesting findings in this paper. First, they found that all the STACs (STAC1-3) can abolish/reduce the calmodulin (CaM)-mediated CDI of L-type (Ca_V_1 sub-family) Ca_V_s (a finding also recently reported by other groups, e.g. Polster et al., 2015, Campiglio et al., 2018). Second, the authors used the FRET protein-interaction assay and found that STAC3 interacts with the proximal Ca^2+^-inactivating (PCI) segment of the Ca^2+^-inactivating (CI) module in the C-terminus of Ca_V_1.3, but not with the other segments, including the IQ motif immediately C-terminal to the PCI and the II/III loop previously found by others to interact with STACs in the Ca_V_1.1 (Yuen et al., 2017, Polster et al., 2018) or Ca_V_1.2 (Campiglio et al., 2018). Intriguingly, the authors found that a short "linker" between the C1 and the SH3 regions on STAC is required for STAC's suppression of CDI, and infusion of a U-motif peptide derived from this linker region is sufficient to remove CDI. This finding again is apparently contrast with a previous conclusion by others that the C1 domain is the major CDI-slowing determinant.Mechanistically, the authors conclude that STACs allosterically control CDI and "lock" the channel into a high opening probability state, a mechanism contrast to a previously proposed one by which STAC competes with CaM. Interestingly, the authors also discovered that the regulation of Na_V_'s CDI by FGF has a structural requirement analogous to that of the regulation of Ca_V_s by STACs. Perhaps the most remarkable finding in the paper is that the authors were able to introduce an artificial Mona SH3-binding sequence in the PCI region to confer complete CDI inhibition by Mona SH3, a protein that doesn't affect the wild-type Ca_V_. Overall, there are many apparent "discrepancies" between the current paper and the two PNAS papers (Yuen et al., 2017, Campiglio et al., 2018), ranging from the STAC binding sites on Ca_V_s (PCI in this paper vs. the IQ motif and II/III loop in the others), and the important segment on STAC (linker vs. C1) to the mechanism (allosteric vs. competitive). The authors might want to elaborate the differences and provide potential explanations.

We thank the reviewers for this excellent suggestion. We could not include an in-depth discussion on apparent discrepancies in our original submission as multiple papers here were only published after our initial submission (e.g. Campiglio et al., 2018 and Polster et al., 2018). The revised Discussion section now includes a subsection entitled “Relationship to prior studies” to compare and contrast these findings.

4) Writing. A comment on style: The authors use a prose style with so many dramatic statements and intensifying adjectives that it will likely irritate and even alienate some readers. In my opinion, a more sober presentation in a standard style that lets the results speak for themselves is more effective. Insisting on telling the reader how important the results are (like repeatedly using "Remarkably" for results that seem very straightforward, like seeing similar effects on Ca_v_1.4 as for Ca_v_1.1, Ca_v_1.2, and Ca_v_1.3) can make the reader think that he or she is receiving a sales pitch rather than being told results. There are too many examples of over-dramatic language to list but one was the statement that stac and fhf exert "orthogonal" control because one interacts with calcium channels and one with sodium channels. Calmodulin interacts with dozens if not hundreds of proteins, and of course there are ways that the interactions are regulated differently – is each one of these an orthogonal dimension? I had a similar reaction to sentences like "Synergistic analysis of the two proteins delineate mechanisms that confer personalized modulation and inform upon physiological consequences" and "Repurposing this modulatory principle furnishes a general strategy for engineering synthetic modulators that patently switch CaV and NaV channel feedback." (These sentences lie between a lack of clear meaning and inaccuracy.)

We apologize for our enthusiasm. We have revised the manuscript accordingly.

[Editors' note: further revisions were requested prior to acceptance, as described below.]

The manuscript has been improved but there are some important issues that need to be addressed, as outlined below:One reviewer (whose comments are included verbatim below) identified some inconsistencies in the match between figure panels cited in the text and in the figures, and also noted that the control plots in the last two supplementary figures (Figure 7—figure supplement 1 and Figure 8—figure supplement 1) appear to be identical, but the duplication (if these are indeed duplicates) is neither justified nor indicated in the legends. The same panel appears to be duplicated in Figure 1—figure supplement 1 and Figure 3—figure supplement 1, although in the latter it is labeled as N=6 instead of N=9 as in the other three panels. These issues raise concern about the care with which the manuscript was assembled and presented.Before a final decision is rendered, please ensure (1) that the correct data and figure panels are present throughout the manuscript, (2) that they are called correctly, (3) that any duplication of panels to facilitate comparison is clearly indicated as such (with appropriate justification for a common control group), and (4) that there are no other related errors. In your revision, please clearly indicate the corrections that have been made.

We thank all reviewers and editors for the careful evaluation of our manuscript. We sincerely apologize for the errors and inconsistencies in our text and figures. These were inadvertent and have been rectified. The *n* values for each figure were verified/corrected, and the figures are also now referenced correctly.

Figure duplications are elaborated below:

1) Figure 6C control bar is duplicated from Figure 1B. The duplication is stated explicitly and highlighted in yellow.

Justification for duplication *–* in this experiment, we pipette dialyze U-motif into HEK293 cells transfected with Ca_V_1.2 to probe changes in CDI. The appropriate control relation here would be Ca_V_1.2 without the peptide dialysate, which is equivalent to Ca_V_1.2 at baseline. As control experiments in Figure 1A-B also deduced Ca_V_1.2 CDI at baseline (i.e. without stac), we pooled the datasets together.

In the previous submission, Figure 6—figure supplement 2B was duplicated from Figure 1—figure supplement 1A. In the revised we have removed this duplication to minimize any confusion.

2) Control bars in Figures 1D, 7E, and 8C are duplicated to facilitate comparison and for symmetry.

Justification for duplication *–* Figure 1D compares CDI of Ca_V_1.3 in the presence of stac to baseline levels. Figure 7E compares CDI of Ca_V_1.3 with the addition of fhf1 to baseline levels, and Figure 8C compares CDI of Ca_V_1.3 with the addition of Mona SH3 to baseline levels. In all three cases, as the appropriate control relation would be the baseline CDI of Ca_V_1.3 under endogenous levels of CaM and without the addition regulatory protein being tested, we duplicated the control bars.

In the previous submission, Control *r*_300_ relations in Figure 1—figure supplement 1B, Figure 3—figure supplement 1B, Figure 7—figure supplement 7B, and Figure 8—figure supplement 1B were duplicates. We have removed these duplications to avoid any confusion.

3) Control CDI_max_ bar in Figure 7C is duplicated from Figure 1N to facilitate comparison.

Justification for duplication *–* in Figure 1, we compare CDI of Na_V_1.4 at baseline with stac2 overexpression. Similarly, in Figure 7, we compare CDI of Na_V_1.4 at baseline with fhf1b overexpression. The duplication here occurs as the control relation in both cases is CDI of Na_V_1.4 at baseline with addition of an auxiliary regulatory protein.

In the previous submission, the CDI – Ca relationship for Na_V_1.4 at baseline was duplicated in Figure 7—figure supplement 1B from Figure 1—figure supplement 1G. This has been removed from the revised version to minimize confusion.

4) Control traces in Figure 3—figure supplement 1C was duplicated from Figure 3A to preserve symmetry of the figure and for ease of comparison. We left this duplication intact.

Justification for duplication – the supplementary figure like the main text figure shows that CaM fusion protects from dominant negative CaM, a competitive inhibitor of CDI. Figure 3—figure supplement 1C shows exemplar traces in the presence and absence of dominant negative CaM to be able to visually compare kinetics of inactivation. The steady state values are reported in panel D.

Reviewer comments:The revision generally addresses the main issues raised on the original version. The data are extensive and comprise a very comprehensive picture of the molecular interactions with the multiple STAC proteins with multiple calcium channels. It is still unproven that the STAC interaction functions physiologically to "tune" Ca-dependent channel inactivation, but the authors make a good point that testing this via RNAi would be technically challenging given current experimental systems and making knock-out mice would be a major project. The fact that STAC proteins are heavily expressed in the brain gives some credence to the idea that they could be functionally important in neurons, or knock-outs.The authors need to proof-read the manuscript more carefully. The reviewer copy was hard to read, as it consisted of a pdf incorporating cross-outs and additions, which made spotting errors difficult. However, it is obvious that not all changes in re-numbering figure references were done correctly (e.g. in the first paragraph of the subsection “Stac selectively suppresses Ca^2+^-feedback of Ca_V_1 channels”, Figure 2A should now be 1A, and in the second paragraph of the subsection “Stac interacts with CaV1 CI module to elicit CDI suppression”, Figure 3B should be 2B).Also the fact that the figures were not numbered made the reviewing process more difficult than was necessary. Also, at the end of the manuscript there are supplementary figures that are exact duplicates.

We thank the reviewer for these comments and for the careful reading of our manuscript. We also apologize for the confusion. We realized *eLife* requested that we incorporate track changes for resubmission only after we had made substantial textual changes. As such, we attempted to reincorporate our changes that then led to errors. We have corrected these in the revision and have also removed track changes here. We are not sure how to number the figures as we had to submit these separately as individual image files.